# PROXIMAL MAPPING LOSS: UNDERSTANDING LOSS FUNCTIONS IN CROWD COUNTING & LOCALIZATION

**Wei Lin[1], Jia Wan[2] & Antoni B. Chan[1]**
[1]Department of Computer Science, City University of Hong Kong, Hong Kong SAR,
[2]School of Computer Science and Technology, Harbin Institute of Technology, Shenzhen
`elonlin24@gmail.com, jiawan1998@gmail.com, abchan@cityu.edu.hk`

## ABSTRACT

Crowd counting and localization involve extracting the number and distribution of crowds from images or videos using computer vision techniques. Most counting methods are based on density regression and are based on an "intersection" hypothesis, *i.e.*, one pixel is influenced by multiple points in the ground truth, which is inconsistent with reality since one pixel would not contain two objects. This paper proposes Proximal Mapping Loss (PML), a density regression method that eliminates this hypothesis. PML divides the predicted density map into multiple point-neighbor cases through nearest neighbor, and then dynamically constructs a learning target for each sub-case via proximal mapping, leading to more robust and accurate training. Furthermore, PML is theoretically linked to various existing loss functions, such as Gaussian-blurred L2 loss, Bayesian loss, and the training schemes in P2PNet and DMC, demonstrating its versatility and adaptability. Experimentally, PML significantly improves the performance of crowd counting and localization, and illustrates the robustness against annotation noise. The code is available at `https://github.com/Elin24/pml`.

## 1 INTRODUCTION

Automated crowd scene understanding has undergone extensive exploration over the years, primarily due to its broad applicability in video surveillance and public safety services (Wang et al., 2020c; Ma & Chan, 2015). The fundamental task within this research domain is crowd counting (Chan & Vasconcelos, 2009; 2011), wherein statistical data related to crowds, crowd count and distribution, is extracted from images or videos using computer vision technologies. In recent times, the advent of deep learning has significantly enhanced the precision of counting methodologies (Wang et al., 2019; Ma et al., 2019; Lin et al., 2022a; Shu et al., 2022), with some well-performing methods also being applied to crowd instance localization (Song et al., 2021; Wan et al., 2021; Lin & Chan, 2023; Han et al., 2023). Concurrently, diverse domains, including traffic congestion control (Wang et al., 2018; Lu et al., 2018; Lin et al., 2022b) and marine environmental monitoring (Sun et al., 2023), have also benefitted from advances in crowd counting algorithms.

Supervised crowd counting is typically formulated as a regression task. Given a crowd image, the counting model aims to predict a density map whose distribution and count closely align with the ground truth (GT) point map, in which a pixel with a value of one denotes a person's location. A pseudo GT density map is created by convolving a Gaussian kernel with the GT point map (Lempitsky & Zisserman, 2010; Zhang et al., 2016). Subsequently, pixel-wise L2 loss is computed between this intermediate map and predictions to supervise the counting model. In contrast to L2 loss that uses an intermediate GT density map based on the GT point map, Bayesian Loss (BL) (Ma et al., 2019) offers an alternative method for supervision based purely on the GT point map. Specifically, BL computes the posterior probability of each pixel having a GT point to establish an intermediate point map, which is then compared with the GT for supervision. Both models make theoretical sense, and positive results have been obtained through experiments. Although L2 loss and BL are developed based on different principles, Wan et al. (2021) demonstrates that both L2 and BL are special cases of a generalized loss (GL), which computes a transport plan matrix between the predicted density map and GT point map using unbalanced optimal transport (UOT) (Peyré et al., 2019). The transport matrix represents both an intermediate density map, by marginalizing over the columns, or a point map, by marginalizing over the rows, thus providing a theoretical connection between GL

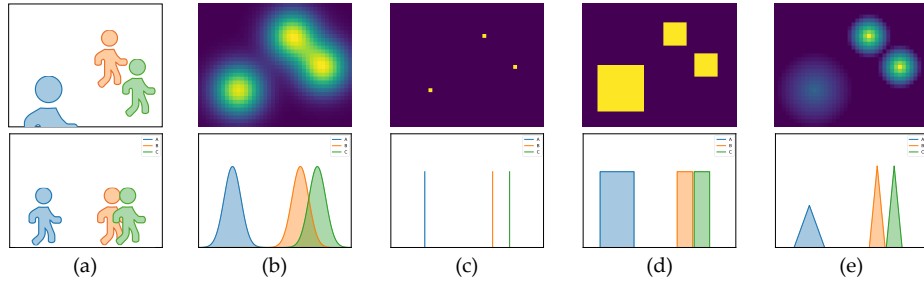

Figure 1: Design principle of counting methods. The top row presents a 2D representation, while the bottom row showcases the corresponding 1D representation. (a) A synthetic input with three humans. (b) Density regression with the intersection hypothesis (using a Gaussian prior) or D2CNet (Cheng et al., 2021), where one pixel may correspond to multiple objects. (c) Point prediction. (d) Head region segmentation. (e) Density regression without the intersection hypothesis (proposed method), where one pixel corresponds to one object. The intersection hypothesis in (b) includes the Gaussian-blurred L2 loss, BL, DMC, and GL. Conversely, (c)-(e) assume no intersections among instances.

and both Gaussian-blurred L2 loss and BL. The effectiveness of GL is also substantiated experimentally, where a GL-trained model estimates more accurate density map than with other losses.

Looking closely at these methods, we note that they are all designed with an *intersection hypothesis*: the value of a foreground pixel in the defined learning objective is influenced by multiple instances, *i.e.*, the mixed Gaussian kernel in Fig. 1(a). However, methods based on point detection, such as P2PNet (Song et al., 2021) and PET (Liu et al., 2023), have shown that it is possible to represent a pedestrian's location with just one pixel, as shown in Fig. 1(c). Additionally, LSC-CNN (Sam et al., 2020), TopoCount (Abousamra et al., 2021), and IIM (Gao et al., 2020) localize and count people by clearly defining the boundaries of head regions around each pedestrian in training and then segmenting them out during inference (Fig. 1(d)). iKNN (Olmschenk et al.) and FIDTM (Liang et al., 2022b) localize crowds by detecting the local maximum of predicted inverse distance transform maps (IDTM) (Fig. 1(e)). These methods assume there is no interaction between pixels of different pedestrians, but achieve counting results comparable to those of density-regression models, suggesting that the intersection hypothesis may not be necessary. Moreover, not considering the intersection hypothesis leads to better localization performance, because there are clear boundaries between each instance (Fig. 1(c-e)). In contrast, the intersection hypothesis causes the predicted density maps to overlap (Fig. 1(b)), making it difficult to define clear boundaries between instances and to identify each independent individual, especially in dense scenarios where the density representation of crowds blends together. D2CNet (Cheng et al., 2021) also follows Fig. 1(b), but uses the `max` rather than the `sum` operator when embedding Gaussian kernels to each GT point when constructing the learning objective, which learns to output overlapping probability maps for localization.

Premised on the above successful experience, we argue for eliminating the intersection hypothesis in the density regression framework, so that a model can be trained to achieve both advanced counting and localization performance. Specifically, we propose to divide the predicted density map into multiple non-overlapping irregular patches and then process each patch independently when computing the loss value, following the philosophy of divide and conquer. The dividing stage is implemented via nearest neighbor assignment, which assigns each pixel in the density map to its nearest point in the ground truth. Consequently, in the conquer stage, we only need to compute the loss within each point-neighbor case separately and then sum these values to obtain the final loss value. However, new obstacles emerge when processing the sub-problems after assignment: these point-neighbor cases vary in the number of neighbors, which introduces difficulty in training with constant targets like Gaussian-kernel density maps (Lempitsky & Zisserman, 2010). This is because there is no effective way to determine hyper-parameters, *i.e.*, the variance in the Gaussian distribution. To address this issue, we propose the Proximal Mapping Loss (PML), which uses proximal mapping (Drusvyatskiy, 2017) to dynamically construct a learning target for each sample during training. This target lies between the predicted distribution and the delta function established by the GT point as parameters. Several experimental results illustrate that our PML outperforms most current approaches, exhibiting superior accuracy in crowd count estimation and capturing crowd distributions.

In conjunction with our empirical advancements, we find that the derived PML theoretically encompasses many existing loss functions when processing the point-neighbor case. Specifically, it

can be transformed into the Gaussian-blurred L2 loss by incorporating certain constraints. PML can also be instantiated as a dynamic L2 loss for the counting model, sharing a similar formulation with BL (Ma et al., 2019). Moreover, by restricting the space of the learning objective to the delta function, we derive the matching strategy for the Hungarian algorithm used in P2PNet (Song et al., 2021). By modifying a regularizer in our PML, the loss function based on balanced optimal transport in DM-Count (DMC) (Wang et al., 2020a) also be derived.

To summarize, the contributions of this paper are three-fold:

1. We introduce *Proximal Mapping Loss* (PML), which can effectively train a crowd counting model based on density regression without the intersection hypothesis. It follows the philosophy of divide and conquer — dividing the training sample into many point-neighbor cases and then computing sub-loss by comparing the difference between the prediction and a dynamic learning target defined via proximal mapping.
2. Theoretically, we establish the connection between PML and several widely-used counting methods, including the Gaussian-blurred L2 loss, Bayesian loss (Ma et al., 2019), and the training schemes in P2PNet (Song et al., 2021) and DMC (Wang et al., 2020a).
3. Empirically, we demonstrate the effectiveness of the proposed PML in crowd counting and localization. The counting performance is significantly improved compared to previous models, and the localization results outperform all previous methods.

## 2 RELATED WORK

The design of crowd counting methods primarily focuses on the model structure, loss function for training, and localization processing.

**Crowd Counting with intersection hypothesis.** Most crowd counting methods are trained with the Gaussian-blurred L2 loss and follows the scheme of density regression. Early publications such as MCNN (Zhang et al., 2016), Switch CNN (Babu Sam et al., 2017), CSRNet (Li et al., 2018), RAZNet (Liu et al., 2019) focus on addressing scale information in crowds by integrating information from CNN modules with different receptive fields. In contrast, later approaches (Gao et al., 2019; Shi et al., 2019; Yang et al., 2020) aim to extract perspective information from the input image and subsequently allocate detailed attention to regions with dense crowds. MAN (Lin et al., 2022a) introduces an effective approach to embed transformers (Vaswani et al., 2017) into crowd counters, enhancing counting performance.

Instead of designing network modules to enhance the effectiveness of crowd counters, Ma et al. (2019); Wan et al. (2021); Wang et al. (2020a); Ma et al. (2021) focus on training counting model with loss functions based on different principles. Typically, most counting models are trained with Gaussian-blurred L2 loss following MCNN (Zhang et al., 2016), using GT density maps obtained by convolving a Gaussian kernel with GT point maps. In contrast, BL (Ma et al., 2019) directly considers these point maps as priors, computing the posterior probability of density pixels corresponding to the points, leading to a more accurate prediction. Subsequently, DMC (Wang et al., 2020a) introduces balanced optimal transport (OT) to establish the mapping relations between GT points and predicted pixels, while UOTCC (Ma et al., 2021) and GL (Wan et al., 2021) utilize unbalanced OT (UOT). Notably, GL demonstrates that BL and Gaussian-blurred L2 loss are special cases of GL by substituting a half Sinkhorn iteration, providing solid theoretical contributions. Although the Gaussian-based pseudo density map is later replaced by posterior probability or optimal transport, these methods for density regression still operate under the intersection hypothesis. There are no works that design a density regression method without the intersection hypothesis. In D2CNet (Cheng et al., 2021), the learning objective is a probability map, where each pixel's value is the output of a Gaussian function with the distance to its nearest GT point as the argument. This approach differs from the use of density maps in constructing the learning objective, as it uses the `max` operator instead of the `sum` operator to embed Gaussian kernels at each GT point's location.

**Crowd Counting without intersection hypothesis.** Frameworks that do not use the intersection hypothesis, like point detection, head segmentation, and inverse distance transform map (IDTM) regression, have also been explored. P2PNet (Song et al., 2021) is the first one proposing to count and localize pedestrians via point detection, treating each pedestrian as a point and predicting their locations through a single-stage detection framework. This approach distinguishes itself from pre-

vious density-based methods by directly acquiring localization information. Follow-up works, such as CLTR (Liang et al., 2022a) and PET (Liu et al., 2023), adopt a similar strategy for predicting points but employ distinct network structures. Apart from these methods, head detection is another popular trend for counting without the intersection hypothesis. LSC-CNN (Sam et al., 2020) and TopoCount (Abousamra et al., 2021) use pseudo and coarse box annotations to supervise the detection network, while IIM (Gao et al., 2020) adopts accurate box annotations provided in NWPU-Crowd (Wang et al., 2020c) to train the counting model. IDTM is also investigated for counting without the intersection hypothesis. FIDTM Liang et al. (2022b) and iKNN (Olmschenk et al.) define the learning objective of each pixel as the inverse value of the distance to the k-nearest point in the ground truth, and locate pedestrians by finding peak positions in a $3 \times 3$ patch during inference.

The above methods without the intersection hypothesis demonstrate outstanding localization performance, but their counting performance is not as good as density regression methods. However, the above methods have inspired us to design a density regression method without the intersection hypothesis. We show that our proposed method encompasses many existing crowd counting loss functions, including BL (Ma et al., 2019) and the widely-used Gaussian-blurred L2 loss, as well as the training schemes presented in P2PNet (Song et al., 2021) and DMC (Wang et al., 2020a).

## 3  PROXIMAL MAPPING LOSS (PML)

To remove the intersection hypothesis existing in current density regression methods, we adopt nearest neighbors to split the density map into multiple irregular patches without overlapping. In this way, the loss computation is divided into multiple simpler sub-problems, as each point-neighbor case is handled independently. For each case, we compute the loss in a dynamic way based on proximal mapping. Overall, PML is based on the principle of divide and conquer.

### 3.1  DIVIDE STAGE: NEAREST NEIGHBOR

Mathematically, we represent the predicted density map with $n$ pixels as $\mathcal{A} = \{(a_i, \boldsymbol{x}_i)\}_{i=1}^n$, in which $a_i \in \mathbb{R}_+$ represents the predicted density value at pixel located at $\boldsymbol{x}_i \in \mathbb{R}^2$. Similarly, the corresponding ground truth with $m$ pedestrians is represented as $\mathcal{B} = \{(1, \boldsymbol{y}_j)\}_{j=1}^m$, with $\boldsymbol{y}_j \in \mathbb{R}^2$ represents the GT location of the $j$-th annotation. The "1" in each element of $\mathcal{B}$ implicates that the count equals to 1 for each point in GT. Accordingly, the divide stage in PML is formulated as:

$$\mathcal{L}(\mathcal{A}, \mathcal{B}) = \sum\nolimits_{j=1}^m \tilde{\mathcal{L}}(\tilde{\mathcal{A}}_j, \boldsymbol{b}_j), \quad \tilde{\mathcal{A}}_j = \{(a_i, \boldsymbol{x}_i)\}_{i \in \mathcal{X}_j} \quad \boldsymbol{b}_j = (1, \boldsymbol{y}_j), \tag{1}$$

$$\mathcal{X}_j = \{ i \mid \|\boldsymbol{x}_i - \boldsymbol{y}_j\|_2 \le \|\boldsymbol{x}_i - \boldsymbol{y}_k\|_2, \ \forall \boldsymbol{y}_k \in \mathcal{B} \}, \tag{2}$$

where $\mathcal{X}_j$ collects the indices of all nearest neighboring pixels to $\boldsymbol{y}_j$, and $\tilde{\mathcal{A}}_j$ represents the corresponding density pixel set. The divide stage splits the construction of loss value $\mathcal{L}(\mathcal{A}, \mathcal{B})$ into $m$ independent sub-loss functions. Thus, we only have to analyze any one of these sub-problems, $\mathcal{L}(\tilde{\mathcal{A}}_j, \boldsymbol{b}_j)$, to ease the theoretical analysis.

### 3.2  CONQUER STAGE: PROXIMAL MAPPING OF TRANSPORT COST

We next conquer the sub-problem $\mathcal{L}(\tilde{\mathcal{A}}_j, \boldsymbol{b}_j)$ presented in (1). To simplify the notation, we remove the subscript $j$ – we represent the GT location as $\boldsymbol{y} \in \mathbb{R}^2$, corresponding to $\boldsymbol{b}_j$. We assume that there are $\tilde{n}$ neighbor pixels collected in $\tilde{\mathcal{A}}_j$, which we denote as $\tilde{\mathcal{A}} = \{(a_i, \boldsymbol{x}_i)\}_{i=1}^{\tilde{n}}$. Defining $\boldsymbol{a} = [a_i]_{i=1}^{\tilde{n}}$, we construct the learning objective by minimizing the transport cost $f(\boldsymbol{a}) = \boldsymbol{c}^\top \boldsymbol{a}$, where $\boldsymbol{c} = [c_i]_{i=1}^{\tilde{n}}$ measures the cost when moving a unit mass from $\boldsymbol{x}_i$ to $\boldsymbol{y}$, which is commonly defined as Euclidean distance $\|\boldsymbol{x}_i - \boldsymbol{y}\|_2$ or its exponential (e.g., in GL (Wan et al., 2021)). Here we consider the objective $f(\boldsymbol{a})$ from the perspective of proximal gradient descent. Specifically, after the $t$-th round of parameter updates, the new prediction, $\boldsymbol{a}_{t+1}$, of the counting network should be close to the *proximal mapping* (Parikh et al., 2014) of the previous $\boldsymbol{a}_t$:

$$\boldsymbol{a}_{t+1} \approx \underset{\boldsymbol{p}}{\arg\min} \ \underbrace{f(\boldsymbol{a}_t) + \nabla f(\boldsymbol{a}_t)^\top (\boldsymbol{p} - \boldsymbol{a}_t)}_{\text{linear approximation of } f(\boldsymbol{a}_{t+1})} + \underbrace{\frac{\tau}{2}\|\boldsymbol{p} - \boldsymbol{a}_t\|^2}_{\text{regularizer}}, \tag{3}$$

where $\tau$ is a hyperparameter controlling the size of the neighborhood for mapping, and $\nabla f(\boldsymbol{a}_t) = \boldsymbol{c}$ is the gradient *w.r.t* $\boldsymbol{a}_t$. The regularizer in (3) is required to prevent $\boldsymbol{a}_{t+1}$ from straying far from $\boldsymbol{a}_t$

| loss function | $\tau$ | $\mathcal{D}_\varphi(\boldsymbol{p}, \boldsymbol{a})$ | $\xi$ | $\boldsymbol{p}^*$ |
|---|---|---|---|---|
| L2 loss (Zhang et al., 2016) | 0 | $\frac{1}{2}\|\boldsymbol{p}-\boldsymbol{a}\|^2$ | $\mathcal{N}(\mu|\Sigma)$ | $\mathcal{N}(0|\sigma\mathbf{1}_{2\times 2})$ |
| Bayesian loss (Ma et al., 2019) | $\frac{1}{|\mathbf{1}^\top\boldsymbol{a}-1|}$ | $\frac{1}{2}\|\boldsymbol{p}-\boldsymbol{a}\|^2$ | - | $\boldsymbol{a}-\frac{1}{|\mathbf{1}^\top\boldsymbol{a}-1|}\boldsymbol{c}+\eta$ |
| P2PNet (Song et al., 2021) | - | $\|\boldsymbol{p}-\boldsymbol{a}\|_1$ | $\delta(\cdot)$ | $\delta(\arg\min_j \boldsymbol{c}_j - \tau\boldsymbol{a}_j)$ |
| DM-Count (Wang et al., 2020a) | $\infty$ | $\mathtt{KL}(\boldsymbol{p} \mid \boldsymbol{a})$ | - | $\boldsymbol{a}/\|\boldsymbol{a}\|_1$ |

Table 1: Different crowd counting loss functions implemented from PML. $\tau$ and $\mathcal{D}_\varphi(\boldsymbol{p}, \boldsymbol{a})$ are defined in (5). $\xi$ and $\boldsymbol{p}^*$ are the distribution formula and the analytical solution of learning objective.

since the linear approximation only works well in the neighborhood of the current $\boldsymbol{a}_t$. Compared with directly computing the loss between prediction and point GT, the proximal mapping in (3) points out a *reachable* and *soft* target for optimization since the estimated $\boldsymbol{a}_{t+1}$ is an intermediate learning object between the current prediction $\boldsymbol{a}_t$ and GT. Thus, we can design a loss function according to (3) by adding a constraint that $\boldsymbol{p}$ sum to 1, and dropping terms that do not depend on $\boldsymbol{p}$:

$$\mathcal{L}(\tilde{\mathcal{A}}, \boldsymbol{b}) = \min_{\boldsymbol{p}\in\xi} \boldsymbol{c}^\top\boldsymbol{p} + \frac{\tau}{2}\|\boldsymbol{p}-\boldsymbol{a}\|^2, \tag{4}$$

where $\xi \subseteq \{\boldsymbol{p} \mid \boldsymbol{p}^\top\mathbf{1} = 1, \boldsymbol{p} \in \mathbb{R}^{\tilde{n}}\}$, and $\nabla f(\boldsymbol{a}_t)$ is replaced by its value $\boldsymbol{c}$. Furthermore, the squared L2-norm can be replaced with other distances, e.g., *Bregman divergence*, to achieve different implementations. Therefore, the general formulation for PML is:

$$\mathcal{L}(\tilde{\mathcal{A}}, \boldsymbol{b}) = \min_{\boldsymbol{p}\in\xi} \boldsymbol{p}^\top\boldsymbol{a} + \tau\mathcal{D}_\varphi(\boldsymbol{p}, \boldsymbol{a}), \tag{5}$$

where $\mathcal{D}_\varphi(\boldsymbol{p}, \boldsymbol{a})$ is the Bregman divergence w.r.t. a strictly convex function $\varphi$. Note that the squared L2-norm is a special case: $\mathcal{D}_\varphi = \frac{1}{2}\|\boldsymbol{p}-\boldsymbol{a}\|^2$ when $\varphi(\cdot) = \frac{1}{2}\|\cdot\|^2$.

## 4 RELATIONSHIP BETWEEN PML AND OTHER LOSS FUNCTIONS

In this section, we show that PML encompasses previous loss functions, including L2 loss, BL (Ma et al., 2019), P2PNet (Song et al., 2021), and DMC (Wang et al., 2020a).

### 4.1 FROM DYNAMIC L2 LOSS TO GAUSSIAN-BLURRED L2 LOSS

We derive the L2 loss from PML with $\varphi(\cdot) = \frac{1}{2}\|\cdot\|^2$, i.e., Eq. 4. Specifically, (4) is rewritten as:

$$\mathcal{L}_2 = \min_{\boldsymbol{p}} \boldsymbol{c}^\top\boldsymbol{p} + \frac{\tau}{2}\|\boldsymbol{p}-\boldsymbol{a}\|_2^2, \quad s.t. \quad \boldsymbol{p}^\top\mathbf{1} = \sum_{i=1}^{\tilde{n}} p_i = 1. \tag{6}$$

Applying Lagrange multipliers, the analytical solution $\boldsymbol{p}^*$ can be derived as

$$\boldsymbol{p}^* = \boldsymbol{a} - \frac{1}{\tau}\boldsymbol{c} + \eta, \qquad \eta = \frac{1}{\tilde{n}}\left[1 - \left(\boldsymbol{a} - \frac{1}{\tau}\boldsymbol{c}\right)^\top\mathbf{1}\right]. \tag{7}$$

Note that $\eta$ takes the role as "filler" such that $\boldsymbol{p}^{*\top}\mathbf{1} = 1$. Substituting (7) into (6) and taking the derivative of $\mathcal{L}_2$ with respect to $\boldsymbol{a}$:

$$\frac{\partial\mathcal{L}_2}{\partial a_i} = c_i - \tau\eta \quad \Rightarrow \quad \mathcal{L}_2 = \frac{\tau}{2}\|\boldsymbol{a} - \mathtt{detach}(\boldsymbol{p}^*)\|_2^2. \tag{8}$$

$\boldsymbol{p}^*$ in (8) is detached from the computation graph in implementation, and the RHS is the loss function for training a crowd counting network.

From (6) and (7), we note that $\boldsymbol{p}^*$ is an intermediate distribution between $\sigma(\boldsymbol{y}_j)$ and $\tilde{\mathcal{A}}$, and $\tau$ controls how close $\boldsymbol{p}^*$ is to the latter. In extreme cases, we have $\lim_{\tau\to 0} \boldsymbol{p}^* = \delta(\boldsymbol{y})$ and $\lim_{\tau\to\infty} \boldsymbol{p}^* = \boldsymbol{a} + (1 - \mathbf{1}^\top\boldsymbol{a})$. However, traditional L2 loss is likely to formulate $\boldsymbol{p}^*$ as a Gaussian distribution, which can be achieved by limiting $\xi$ within a Gaussian prior, i.e., $\xi \subseteq \mathcal{N}(\boldsymbol{\mu}|\Sigma)$, or projecting $\boldsymbol{p}^*$ into the feasible region after obtaining $\boldsymbol{p}^*$:

$$\boldsymbol{p}^{*\prime} \leftarrow \mathcal{N}(\boldsymbol{\mu} \mid (\boldsymbol{x} - \boldsymbol{\mu})^\top(\boldsymbol{x} - \boldsymbol{\mu})), \qquad \boldsymbol{\mu} = \boldsymbol{x}^\top\boldsymbol{p}^*. \tag{9}$$

Note that (9) is still a dynamic solution depending on $\tilde{\mathcal{A}}$. Fitting the original Gaussian-blurred L2 loss tightly, i.e., removing the dynamic property, can be achieved by setting $\tau = 0$ so that $\boldsymbol{p}^* = \delta(\boldsymbol{y})$, and limiting $\Sigma$ with a lower bound (i.e., $\Sigma \succeq \sigma^2\mathbf{I}_{2\times 2}$): $\boldsymbol{p}^{*\prime} \leftarrow \mathcal{N}(\boldsymbol{y}|\sigma^2\mathbf{I}_{2\times 2})$.

However, Compared with fixed Gaussian kernels, adapting the learning target (*e.g.*, the intermediate representation between the density map and ground truth) can lead to better counting performance by better fitting the properties of the data, as explored in numerous studies (Bai et al., 2020; Wan & Chan, 2019; Wan et al., 2020; 2021; Song et al., 2021).

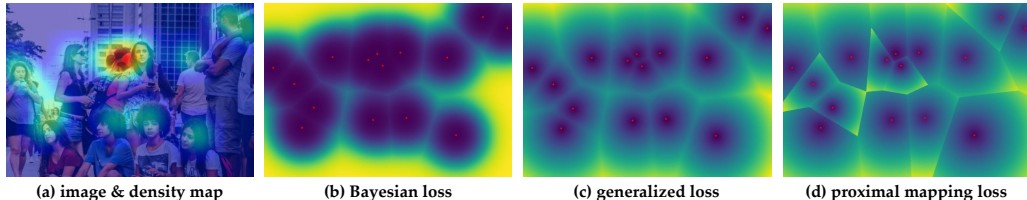

| (a) image & density map | (b) Bayesian loss | (c) generalized loss | (d) proximal mapping loss |

Figure 2: Visualization of the weight item of the background loss in (b) Bayesian loss (Ma et al., 2019), (c) generalized loss (Wan et al., 2021), and (d) our proximal mapping loss. (a) displays the predicted density map overlayed on the input image. The weight map of PML has clearer boundaries of instances, thus resulting in better localization results.

## 4.2 RELATIONSHIP WITH BAYESIAN LOSS

Bayesian loss (Ma et al., 2019) is formulated as:

$$\mathcal{L}_{Bayesian} = \sum_{i=1}^{n} q(\boldsymbol{y}_0|\boldsymbol{x}_i)a_i + \sum_{j=1}^{m} \left| \sum_{i=1}^{n} q(\boldsymbol{y}_j|\boldsymbol{x}_i)a_i - 1 \right|, \qquad (10)$$

where $q(\boldsymbol{y}_j|\boldsymbol{x}_i)$ with $j > 0$ represents the posterior probability of $\boldsymbol{x}_i$ having the label $\boldsymbol{y}_j$, while $q(\boldsymbol{y}_0|\boldsymbol{x}_i)$ models the background pixels. In the point-neighbor case, i.e., $m = 1$, the posterior probability $q(\boldsymbol{y}_1|\boldsymbol{x}_i) = 1$ since all predicted density values $a_i$ are assigned to the same target $\boldsymbol{y}_1$. Therefore, (10) is simplified as:

$$\mathcal{L}_b = \underbrace{\boldsymbol{q}^\top \boldsymbol{a}}_{\text{background loss}} + \underbrace{\left|\boldsymbol{a}^\top \mathbf{1} - 1\right|}_{\text{count loss}}, \qquad (11)$$

in which $\boldsymbol{q} = [q(\boldsymbol{y}_0|\boldsymbol{x}_i)]_{i=1}^{\tilde{n}}$. Similar to the previous section, the density map is $\tilde{\mathcal{A}} = \sum_{i=1}^{\tilde{n}} a_i\delta(\boldsymbol{x}_i)$, and $\boldsymbol{a} = [a_i]_{i=1}^{\tilde{n}}$. In BL, the absolute term in (11) is called the *count loss*, which forces the sum of $\boldsymbol{a}$ to be close to 1, while the *background loss* $\boldsymbol{q}^\top \boldsymbol{a}$ forces the distribution of $\boldsymbol{a}$ to be close to $\sigma(\boldsymbol{y})$. This is achieved by backpropagating greater gradients to pixels in $\tilde{\mathcal{A}}$ that are far from $\boldsymbol{y}$. Specifically, the $i$-th element $q_i$ is defined based on Bayes' rule with the Gaussian distribution as the prior:

$$q_i = q(\boldsymbol{y}_0|\boldsymbol{x}_i) = \frac{q(\boldsymbol{x}_i|\boldsymbol{y}_0)}{q(\boldsymbol{x}_i|\boldsymbol{y}_0)+1}, \qquad q(\boldsymbol{x}_i|\boldsymbol{y}_0) \stackrel{\text{def.}}{=\!=} \frac{1}{\sqrt{2\pi}\sigma} \exp\left[-\frac{(d-c_i)^2}{2\sigma^2}\right], \qquad (12)$$

where $d = \max_i c_i$. The LHS of (12) shows $\boldsymbol{q}_{[i]} \propto q(\boldsymbol{x}_i|\boldsymbol{y}_0)$, while the RHS holds $q(\boldsymbol{x}_i|\boldsymbol{y}_0) \propto c_i$. Thus the order-preserving is transitive: $\boldsymbol{q} \propto \boldsymbol{c}$, revealing that there must be $\boldsymbol{q}_{[i]} > \boldsymbol{q}_{[j]}$ if $\boldsymbol{c}_i > \boldsymbol{c}_j$.

The relationship between (11) and (4) is not apparent, but there is a connection between BL and the analytical solution of dynamic L2 loss presented in (8), which means both L2 and BL instantiate $\mathcal{D}_\varphi(\boldsymbol{p}, \boldsymbol{a})$ with the L2-norm. Borrowing the conclusion from (8) and substituting $\eta$ into (7), we have:

$$\frac{\partial \mathcal{L}_2}{\partial a_i} = c_i - \frac{1}{\tilde{n}}\boldsymbol{c}^\top \mathbf{1} + \frac{\tau}{\tilde{n}}(\boldsymbol{a}^\top \mathbf{1} - 1) \iff \mathcal{L}_2 = \underbrace{(\boldsymbol{c} - \bar{c})^\top \boldsymbol{a}}_{\text{background loss}} + \underbrace{\frac{\tau}{2\tilde{n}}(\boldsymbol{a}^\top \mathbf{1} - 1)^2}_{\text{count loss}}, \qquad (13)$$

where $\bar{c} = \frac{1}{\tilde{n}}\mathbf{1}^\top \boldsymbol{c} = \frac{1}{\tilde{n}}\sum_{i=1}^{\tilde{n}} c_i$ is the average value of elements in $\boldsymbol{c}$. Without loss of generality, the squared item in (13) could be considered as the *count loss* to force the sum of $\boldsymbol{a}$ close to 1. Meanwhile, $(\boldsymbol{c} - \bar{c})^\top \boldsymbol{a}$ is the *background loss* since greater $c_i$ must result in greater gradient $c_i - \bar{c}$ for $a_i$ via the background loss item. Fig. 2 demonstrates the weight item, *i.e.*, the multiplier of $\boldsymbol{a}$ in the background loss part of BL, GL, and our PML. There is no big difference apart from the sharp boundaries present with PML, which are due to the nearest neighbor regions.

Note that the L2-norm in (13) can be transformed into the L1-norm by setting $\tau$ dynamically according to the absolute error, i.e., $\tau = \texttt{detach}\left(2/|\mathbf{1}^\top \boldsymbol{a} - 1|\right)$, which results in a new solution:

$$\mathcal{L}'_{Bayesian} = (\boldsymbol{c} - \bar{c})\boldsymbol{a} + \frac{1}{\tilde{n}}\left|\mathbf{1}^\top \boldsymbol{a} - 1\right| \iff \boldsymbol{p}^* = \boldsymbol{a} - \left(\frac{1}{2}|\mathbf{1}^\top \boldsymbol{a} - 1|\right)\boldsymbol{c} + \eta. \qquad (14)$$

(14) shows that using L1-norm as the count loss is the same as letting $\boldsymbol{p}^*$ change according to a dynamic $\tau$. When the predicted count is close to 1, $\boldsymbol{p}^*$ is also close to the distribution of $\boldsymbol{a}$, which nearly has no contribution to parameter updates. On the contrary, $\boldsymbol{p}^*$ will be close to $\delta(\boldsymbol{y})$ if the

predicted count is far from GT. Therefore, using the L1-norm is an implicit way to set a dynamic $\tau$ that is correlated with the absolute difference between predicted and GT count. This observation explains the reason why the L1-norm performs better than the L2-norm while playing the role of counting loss in both BL and GL – a dynamic $\tau$ is robust and able to mitigate noise in annotation. Later, we will also show how the L1- and L2-norms affect our PML in experiments.

Despite the similarities between PML and BL, differences also exist. Specifically, $\boldsymbol{q}$ in (11) is constructed using only positive numbers. Therefore, the background loss in BL is expected to reduce all density values, regardless of whether they belong to a pedestrian or the background. In contrast, $\boldsymbol{c} - \bar{c}$ allocates positive weights to pixels that are close to GT points, as long as their distance $c_i$ is smaller than the average distance $\bar{c}$. In other words, the background loss in PML segments the density map into "foreground" and "background," expecting density pixels in the foreground to be promoted while background parts are lowered. This is in contrast to BL, which considers all pixels to belong to the background and lowers their values.

## 4.3 Relationship with P2PNet

Here we build the connection between (5) and the training scheme in P2PNet (Song et al., 2021). We instantiate $\mathcal{D}_\varphi$ with L1-norm, and limit $\boldsymbol{p}$ to a subspace of $\xi$ : $\xi_{p2p} = \{\delta(\boldsymbol{x}_i) | \boldsymbol{x}_i \in \tilde{\mathcal{A}}\}$. Accordingly, (4) can be instantiated as:

$$\mathcal{L}_{p2p} = \min_{\boldsymbol{p} \in \xi_{p2p}} \boldsymbol{c}^\top \boldsymbol{p} + \tau \|\boldsymbol{p} - \boldsymbol{a}\|_1, \qquad \xi_{p2p} = \{\delta(\boldsymbol{x}_i) | \boldsymbol{x}_i \in \tilde{\mathcal{A}}\}, \qquad (15)$$

which can be rewritten as:

$$\mathcal{L}_{p2p} = \min_{\boldsymbol{p} \in \xi_{p2p}} \boldsymbol{c}^\top \boldsymbol{p} + \tau \left[ \boldsymbol{p}^\top (1 - \boldsymbol{a}) + (1 - \boldsymbol{p})^\top \boldsymbol{a} \right] \qquad (16)$$

$$= \min_{\boldsymbol{p} \in \xi_{p2p}} \underbrace{(\boldsymbol{c} - 2\tau \boldsymbol{a})^\top \boldsymbol{p}}_{\text{matching strategy}} + \underbrace{\tau (\boldsymbol{p}^\top 1 + \boldsymbol{a}^\top 1)}_{\text{constant } (1 + \|\boldsymbol{a}\|_1)}, \qquad (17)$$

whose solution is $\boldsymbol{p}^* = \delta(\arg\min_i c_i - 2\tau a_i)$, which is exactly the way to compute the cost matrix for the Hungarian algorithm in P2PNet when defining the $i$-th element in $\boldsymbol{c}$ as $c_i = \|\boldsymbol{x}_i - \boldsymbol{y}\|_2$:

$$\mathcal{C}(\mathcal{A}, \mathcal{B}) = [\tau' \|\boldsymbol{x}_i - \boldsymbol{y}_j\|_2 - \boldsymbol{a}]_{i \in N, j \in M}. \qquad (18)$$

By setting $\tau = \frac{1}{2\tau'}$, the matching strategy in (17) is the same as (18).

## 4.4 Relationship with DM-Count (DMC)

DMC (Wang et al., 2020a) is another approach to compute the loss for crowd counting models. In contrast to GL, DMC uses balanced optimal transport to match normalized predictions and normalized GT. In the point-neighbor case, the DMC loss is simplified as:

$$\mathcal{L}_{dmc} = \underbrace{\boldsymbol{c}^\top \frac{\boldsymbol{a}}{\|\boldsymbol{a}\|_1}}_{\text{OT loss}} + \tau \underbrace{|1^\top \boldsymbol{a} - 1|}_{\text{count loss}}. \qquad (19)$$

The OT loss is simplified in (19) since all normalized mass in $\frac{\boldsymbol{a}}{\|\boldsymbol{a}\|_1}$ are transported to the same GT point $\boldsymbol{y}$. Here we show that DMC is a special case when $\mathcal{D}_\varphi$ is the generalized KL divergence $\text{KL}(\boldsymbol{p}|\boldsymbol{a})$ (Peyré et al., 2019), whose variables are defined on $\mathbb{R}_+^{\tilde{n}}$:

$$\varphi(\boldsymbol{x}) = \boldsymbol{x}^\top \log \boldsymbol{x} - \boldsymbol{x}^\top 1 \quad \Rightarrow \quad \mathcal{D}_\varphi(\boldsymbol{p}, \boldsymbol{a}) = \boldsymbol{p}^\top \log \frac{\boldsymbol{p}}{\boldsymbol{a}} - \boldsymbol{p}^\top 1 + \boldsymbol{a}^\top 1. \qquad (20)$$

By substituting (20) into (5), we have:

$$\mathcal{L}_{dmc} = \min_{\boldsymbol{p}} \boldsymbol{c}^\top \boldsymbol{p} + \tau (\boldsymbol{p}^\top \log \frac{\boldsymbol{p}}{\boldsymbol{a}} - \boldsymbol{p}^\top 1 + \boldsymbol{a}^\top 1) \quad s.t. \quad \boldsymbol{p}^\top 1 = 1, \qquad (21)$$

The close-form solution $\boldsymbol{p}^*$ is obtained by applying Lagrange multipliers to (21), resulting in

$$p_i^* = \frac{a_i \exp(-c_i/\tau)}{\eta}, \qquad \eta = \sum_j a_j \exp(-c_i/\tau). \qquad (22)$$

Here, $\eta$ also serves as "filler", ensuring the sum of elements in $\boldsymbol{p}$ equals 1. As $\tau \to \infty$, $\boldsymbol{p}$ becomes the normalized density map: $\boldsymbol{p} = \frac{\boldsymbol{a}}{\|\boldsymbol{a}\|_1}$. Replacing $\boldsymbol{p}$ with the normalized $\boldsymbol{a}$ in (21), the simplified loss function is:

$$\mathcal{L}_{dmc}^{(\varphi)} = \underbrace{\boldsymbol{c}^\top \frac{\boldsymbol{a}}{\|\boldsymbol{a}\|_1}}_{\text{OT loss}} + \tau \underbrace{\left(1^\top \boldsymbol{a} - \log\left(1^\top \boldsymbol{a}\right) - 1\right)}_{\text{count loss}}. \qquad (23)$$

The OT loss is the same as that in (19), except that the count loss takes a KL divergence formulation: $f(x) = x - \log x - 1$, which is a convex function and reaches the minimum at the solution $x^* = 1$.

| METHOD | (backbone) | ShTech A | | ShTech B | | UCF-QNRF | | JHU ++ | | NWPU | |
|---|---|---|---|---|---|---|---|---|---|---|---|
| | | MAE | MSE | MAE | MSE | MAE | MSE | MAE | MSE | MAE | MSE |
| MCNN (Zhang et al., 2016) | | 232.5 | 714.6 | 110.2 | 173.2 | 277.0 | 426.0 | 188.9 | 483.4 | 232.5 | 714.6 |
| CSRNet (Li et al., 2018) | (VGG-16) | 68.2 | 115.0 | 10.6 | 16.0 | 110.6 | 190.1 | 85.9 | 309.2 | 121.3 | 387.8 |
| SFCN (Wang et al., 2019) | (ResNet-101) | 64.8 | 107.5 | 7.6 | 13.0 | 102.0 | 171.4 | 77.5 | 297.6 | 105.7 | 424.1 |
| BL (Ma et al., 2019) | (VGG-19) | 62.8 | 101.8 | 7.7 | 12.7 | 88.7 | 154.8 | 75.0 | 299.9 | 105.4 | 454.2 |
| KDMG (Wan et al., 2020) | (VGG-19) | 63.8 | 99.2 | 7.8 | 12.7 | 99.5 | 173.0 | 69.7 | 268.3 | 100.5 | 415.5 |
| DMC (Wang et al., 2020a) | (VGG-19) | 59.7 | 95.7 | 7.4 | 11.8 | 85.6 | 148.3 | 68.4 | 283.3 | 88.4 | 357.6 |
| NoiseCC (Wan & Chan, 2020) | (VGG-19) | 61.9 | 99.6 | 7.4 | 11.3 | 85.8 | 150.6 | 67.7 | 258.5 | 96.9 | 534.2 |
| P2PNet (Song et al., 2021) | (VGG-16bn) | 52.7 | 85.6 | 6.3 | 9.9 | 85.3 | 154.5 | - | - | 77.4 | 362.0 |
| UOTCC (Ma et al., 2021) | (VGG-19) | 58.1 | 95.9 | 6.5 | 10.2 | 83.3 | 142.3 | 60.5 | 252.7 | 87.8 | 387.5 |
| GL (Wan et al., 2021) | (VGG-19) | 61.3 | 95.4 | 7.3 | 11.7 | 84.3 | 147.5 | 59.9 | 259.5 | 79.3 | 346.1 |
| ChfL (Shu et al., 2022) | (VGG-19bn) | 57.5 | 94.3 | 6.9 | 11.0 | 80.3 | 137.6 | 57.0 | 235.7 | 76.8 | 343.0 |
| PET (Liu et al., 2023) | (VGG-16bn) | **49.3** | **78.8** | 6.2 | 9.7 | 79.5 | 144.3 | 58.5 | 238.0 | 74.4 | 328.5 |
| STEERER (Han et al., 2023) | (HRNet) | 54.5 | 86.9 | _5.8_ | _8.5_ | _74.3_ | _128.3_ | _54.3_ | 238.1 | **63.7** | _309.3_ |
| PML (ours) | (VGG-16bn) | _50.6_ | _80.7_ | 6.1 | 9.7 | 79.5 | 142.7 | 58.9 | 249.6 | 75.7 | 353.1 |
| PML (ours) | (VGG-19) | 55.5 | 89.0 | 6.0 | 9.3 | 76.6 | 132.2 | 57.4 | **227.4** | 73.6 | 338.6 |
| PML (ours) | (HRNet) | 52.3 | 84.7 | **5.4** | **8.2** | **73.2** | **127.5** | **52.6** | _230.8_ | _63.8_ | **306.9** |

Table 2: Comparison of our PML with recent crowd counting methods.

## 5 EXPERIMENTS

In this section, we present experiments demonstrating the efficacy of our proposed PML in crowd counting and crowd localization.

### 5.1 IMPLEMENTATION DETAILS

In our experiments, nearest neighbor is employed in the divide stage to associate pixels in the density map with its nearest point in the ground truth. During training, the dynamic L2 loss with L1-norm, as described in (14), is utilized to supervise the counting model, showing superior performance. Additionally, a hyperparameter $\epsilon$ is introduced to balance these two terms in PML. Therefore, the final loss is formulated by incorporating (14) into (1) and add $\epsilon$ as the weight:

$$\mathcal{L}(\mathcal{A}, \mathcal{B}) = \boldsymbol{a}^\top \tilde{\mathbf{C}} \mathbf{1} + \epsilon \|\mathbf{P}^\top \boldsymbol{a} - \mathbf{1}\|_1,$$

$$\text{where} \quad \tilde{\mathbf{C}}_{ij} = \mathbf{C}_{ij} - \mathbf{P}_{ij} \frac{\sum_{i=1}^n \mathbf{C}_{ij}}{\sum_{i=1}^n \mathbf{P}_{ij}}, \quad \text{with} \quad \mathbf{C}_{ij} = \frac{\mathbf{P}_{ij} \exp(\|\boldsymbol{x}_i - \boldsymbol{y}_j\|_2)}{\max_i \mathbf{P}_{ij} \exp(\|\boldsymbol{x}_i - \boldsymbol{y}_j\|_2)}. \quad (24)$$

In (24), $\mathbf{P} \in \{0,1\}^{n \times m}$ denotes the matching matrix computed via nearest neighbor, where $\mathbf{P}_{ij} = 1$ if $\|\boldsymbol{x}_i - \boldsymbol{y}_j\|_2 \leq \|\boldsymbol{x}_i - \boldsymbol{y}_k\|_2$ for all $k \leq m$, and $\mathbf{P}_{ij} = 0$ otherwise. Inspired by GL, $\mathbf{C}_{ij}$ is computed by normalizing the exponential Euclidean distance between $\boldsymbol{x}_i$ an $\boldsymbol{y}_j$, in which the normalization is implemented by dividing the maximum distance between the $j$-th target $\boldsymbol{y}_j$ and pixels in its point-neighbor case. Subsequently, subtracting the average distance from $\mathbf{C}_{ij}$ results in $\tilde{\mathbf{C}}_{ij}$ for loss computation, aligning with operation that subtracts the average cost $\bar{c}$ in (14).

The counting model utilizes VGG19 (Simonyan & Zisserman, 2015) and HRNet (Wang et al., 2020b) as the backbone. Details and related ablation study on the structure are illustrated in the appendix.

### 5.2 OVERALL PERFORMANCE ON COUNTING

In Tab. 2, we compare PML with other prominent crowd counting methods. Taking VGG16-bn as backbone, PML achieves similar performance to PET and perform better than P2PNet. Using VGG19 as the backbone, PML achieves significant improvements over BL, DMC, and GL. On the JHU Crowd++ dataset (Sindagi et al., 2020), the MAE of PML closely matches Chfl (57.4 vs. 57.0), while the MSE of PML (227.4) is lower (235.7). When combined with HRNet backbone, PML attains the best performance on ShanghaiTech Part B (Zhang et al., 2016) with an MAE of 5.4 and MSE of 8.2. On the UCF-QNRF dataset (Idrees et al., 2018), the estimation errors (MAE: 73.2, MSE: 127.5) are superior to the latest records set by STEERER (Han et al., 2023) (MAE: 74.3; MSE: 128.3). Additionally, on the larger NWPU benchmark (Wang et al., 2020c), PML demonstrates outstanding performance with an MAE of 63.8 and MSE of 306.9, showcasing its excellent counting performance on large-scale datasets. Note that due to the small size of ShTech A, VGG19 usually overfits and does not achieve good performance, while VGGG16-bn does not overfit and obtains MAE of 50.6 and MSE of 80.7, comparable to PET.

|  | F1-meas. | Prec. | Rec. |
|---|---|---|---|
| RAZNet | 0.599 | 0.666 | 0.543 |
| GL+LM | 0.660 | 0.800 | 0.562 |
| GL+OTM | 0.683 | 0.710 | 0.658 |
| P2PNet | 0.729 | 0.676 | 0.685 |
| PET | 0.742 | 0.752 | 0.732 |
| STEERER+LM | 0.770 | **0.814** | 0.730 |
| PML(VGG-19)+OTM | 0.735 | 0.776 | 0.698 |
| PML(HRNet)+OTM | **0.802** | 0.809 | **0.795** |
| PML(HRNet)+LM | 0.790 | 0.803 | 0.777 |

Table 3: Localization on NWPU-Crowd.

|  | MCNN | | CSRNet | | VGG19 | | HRNet | |
|---|---|---|---|---|---|---|---|---|
|  | MAE | MSE | MAE | MSE | MAE | MSE | MAE | MSE |
| L2 Loss | 186.4 | 283.6 | 110.6 | 190.1 | 98.7 | 176.1 | 92.03 | 157.49 |
| BL | 190.6 | 272.3 | 107.5 | 184.3 | 88.8 | 154.8 | 85.52 | 149.55 |
| NoiseCC | 177.4 | 259.0 | 96.5 | 163.3 | 85.8 | 150.6 | - | - |
| DMC | 176.1 | 263.3 | 103.6 | 180.6 | 85.6 | 148.3 | 82.07 | 144.84 |
| GL | 142.8 | 227.9 | 92.0 | 165.7 | 84.3 | 147.5 | 78.37 | 140.23 |
| GCFL | - | - | 83.0 | 139.8 | 80.3 | 137.6 | - | - |
| PML (ours) | **138.9** | **215.7** | **82.1** | **139.0** | **77.6** | **132.8** | **73.17** | **127.45** |

Table 4: Comparison of loss functions and backbones on UCF-QNRF dataset.

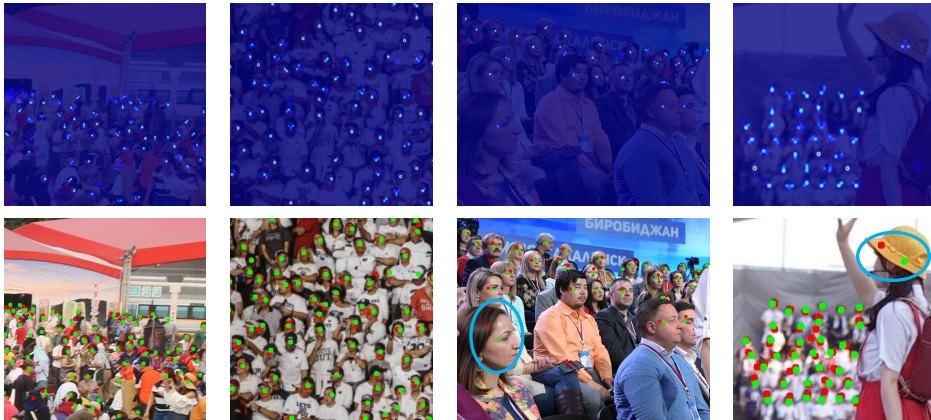

Figure 3: Visualization of samples in the NWPU validation set: density map (top) and estimated point map (bottom) via OTM (Lin & Chan, 2023). In the bottom row, predicted points are annotated in green, while ground-truth points are in red. The blue ellipses highlight examples where PML is robust to annotation noise.

## 5.3 LOCALIZATION PERFORMANCE ON NWPU-CROWD

We applied OTM (Lin & Chan, 2023), which transforms the density map into a point map, to our predictions to localize the people in the crowd, and compared the results with other crowd localization methods, as shown in Tab. 3. PML with VGG19 backbone performs well against GL with OTM (F1: 0.735 *vs.* 0.683), and its F1 is also better than the point-detection-based P2PNet (Song et al., 2021), close to the recent PET (Liu et al., 2023).

PML with HRNet as backbone achieves the best localization performance. Although the precision is a little lower than STEERER (Han et al., 2023), the recall is much higher, resulting a better F1 of 0.802. For fair comparison with STEERER, we also use the local maximum as the localization algorithm. As displayed in the last row of Tab. 3, the performance is still better than STEERER.

In Fig. 3, we present detailed visualizations of predicted density maps and the corresponding point maps estimated via OTM (Lin & Chan, 2023).

## 5.4 COMPARISON WITH OTHER LOSS FUNCTIONS

We applied PML to different counting models and compared when training with different loss functions, including the traditional Gaussian-blurred L2 loss, BL, NoiseCC (Wan & Chan, 2020), a method tackling the problem of noisy annotations, DMC, and GL. Four counting models with different structures and complexities were selected to demonstrate the generality of PML. The test results on UCF-QNRF (Idrees et al., 2018) are presented in Tab. 4. PML effectively trains all these models and further enhances their performance over other loss functions. Compared with GL, PML reduces the MAE of MCNN from 142.8 to 138.9, while MSE is also reduced from 227.9 to 215.7. For CSRNet (Li et al., 2018), VGG19, and HRNet the estimation errors are also considerably reduced. Moreover, our PML outperforms the recent GCFL (Shu et al., 2022; 2023), which transforms the density map into the frequency domain and then designs a generalized characteristic function loss.

## 5.5 THE EFFECT OF COUNT LOSS

Here we discuss the robustness of PML with L1-norm (Eq. 13) as the count loss against annotation noise when compared to L2-norm (Eq. 14). Fig. 4(a)–(b) show the results of adding noise following

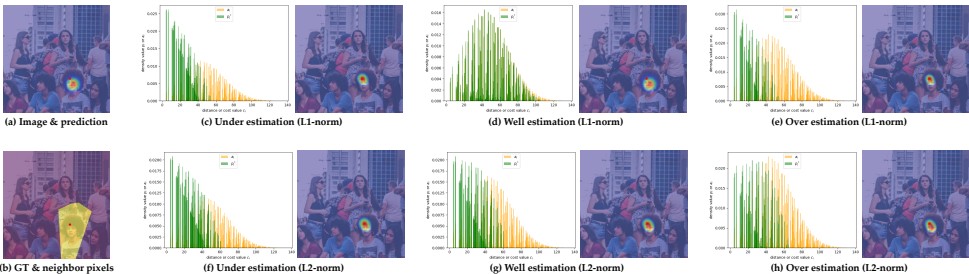

Figure 4: Robustness to annotation error. (a) Adding uniform random noises ranging from a percentage of the image height; (b) Selecting noisy annotations by picking a pixel uniformly within a circle of a specific radius.

Figure 5: Visualization of the learning objective $p^*$ when given noisy annotation and different estimation $a$. (a) shows a prediction that is not aligned with the GT point, and (b) shows the point's neighbor pixels. (c-e) display the distribution of $p^*$ and $a$ with (c) under-estimated, (d) well-estimated, and (e) over-estimated $a$ when L1-norm is applied, while (f-h) display the distribution of L2-norm.

BL Ma et al. (2019) and NCC Wan & Chan (2020). PML with the L1-norm achieves lower estimation errors than the one with the L2-norm across all noise degrees, and outperforms BL and NCC. Additionally, we plot how the MAE changes during training in Fig. 4(c), which shows that training with the L2-norm is unstable, as evidenced by the presence of spurious peaks of high MAE.

The superior performance of L1-norm compared to L2-norm is consistent with the findings in ablation studies of BL (Ma et al., 2019) and GL (Wan et al., 2021). This preference for L1-norm (Eq. 14) may arise from the fact that it defines the hyperparameter $\tau$ as a function of MAE, as opposed to the fixed constant in L2-norm (Eq. 13). This is advantageous for improving robustness since it can mitigate the impact of annotation noise. In Fig. 5, we visualize an example to illustrate this characteristic. When the density map is under- or over-estimated, the learning objective $p^*$ covers most density pixels that are close to the GT point (distance $c_i$ is close to 0) in both L1-norm and L2-norm. However, if $a$ is well-estimated, L1-norm will keep the learning objective $p^*$ close to the estimation (Fig. 5(d)), even if the prediction is far from the annotation. In such cases, the model will consider the GT point as a noisy annotation. In contrast, L2-norm will still attempt to move the prediction closer to the noisy annotation (Fig. 5(g)), which is detrimental to training. Furthermore, we also showcase the PML's robustness in Fig. 3: the two blue ellipses in the last two columns mark examples where predictions are true positives but do not overlap with the GT noisy annotation.

## 6 CONCLUSION

This paper introduces the Proximal Mapping Loss (PML) as an effective training method for crowd counting models. In contrast to previous density regression methods that are based on the intersection hypothesis, PML eliminates it by dividing the predicted density map into multiple point-neighbor cases via nearest neighbors, and then computes the loss value in each sub-case. The loss function is derived by minimizing the difference between current prediction and its proximal mapping. After theoretical analysis, we find that PML has close links with many loss functions for object counting when they are considered in the point-neighbor case, including Gaussian-blurred L2 loss, Bayesian loss, and the training scheme in P2PNet and DMC. In the experiments, PML achieves outstanding performance compared to other counting models and crowd localization methods. Furthermore, different counting models designed with various principles also achieve lower estimation errors when they are trained with PML, providing empirical evidence for the effectiveness of PML.

ACKNOWLEDGEMENTS

This work is supported by a Strategic Research Grant from City University of Hong Kong (Project No. 7005840) and a National Natural Science Foundation of China (Project No. 62406090).

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

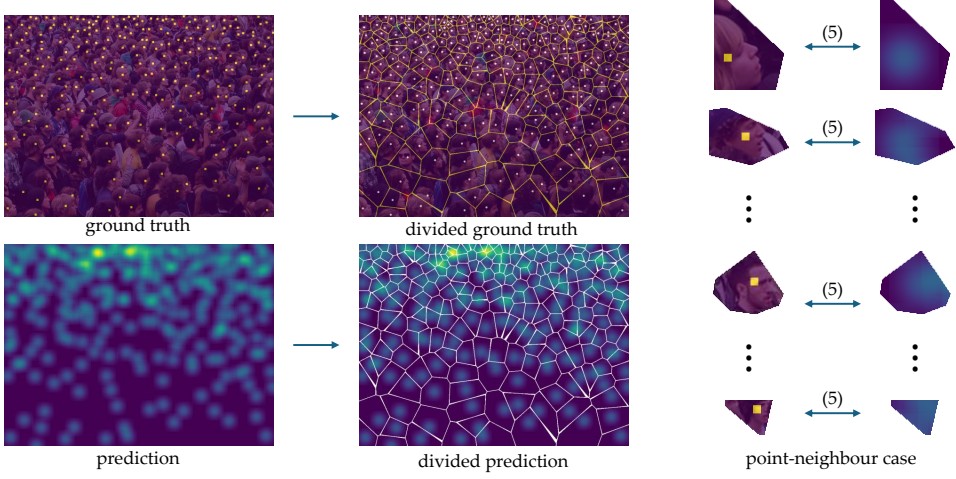

| | |
|---|---|
| **(I) Divide Stage**: assign each pixel to its nearest GT point | **(II) Conquer Stage**: loss computation |

Figure 6: Procedure of the divide and the conquer stage in PML.

## A    OVERVIEW

In this appendix, we discuss more details about the relationship between PML and other loss functions, to show the generalization of PML. Besides, we also present details about the counting model, and more visualization figures are displayed to show the effectiveness of PML.

1. In Sec. B, we present a figure to illustrate how PML is implemented following the principle of divide and conquer;
2. In Sec. C, we demonstrate the difference when PML processes sparse and dense crowd.
3. In Sec. D, an improved P2PNet (Song et al., 2021) is designed from the view of PML and achieves better performance than the vanilla P2PNet;
4. In Sec. E, we discuss the relationship between PML with KL-divergence (21) and DMC (Wang et al., 2020a) in more detail;
5. In Sec. F, experimental results are presented to demonstrate the difference between PML and BL (Ma et al., 2019);
6. In Sec. G, we show that PML is easier to be understand and it is more efficient for training a counting model than GL (Wan et al., 2021).
7. In Sec. H, the model structure used in our experiments is introduced;
8. In Sec. I, ablation studies are presented to discuss issues involving hyper parameters in PML and model structure. Besides, we extend Tab. 4 to more datasets.
9. In Sec. J, the details of results on NWPU benchmark are presented, and some prediction results on NWPU validation set are visualized.

## B    VISUALIZATION OF DIVIDE AND CONQUER

Fig. 6 demonstrates how our PML follows the principle of the divide and conquer. In the (I) divide stage, each pixel in the prediction is assigned to its nearest point in the GT, construct $\mathcal{X}_i$ in (2). After that, each point-neighbor case is conquered independently based on proximal mapping (5).

## C    SPARSE *vs.* DENSE CROWD IN PML

In Fig. 7(a), we plot a figure to show the "average count in each point-neighbor case" *vs.* the "NN distance of each point" for PML, GL, and P2P on the UCF-QNRF dataset, while the bar chart in it demonstrates "the number of point-neighbor cases" *vs.* "NN distance of each point." Here NN distance means the L2 distance between the concerned GT point and its corresponding nearest one in GT, which reflects the sparsity of the crowd near the concerned point. It shows that even the NN

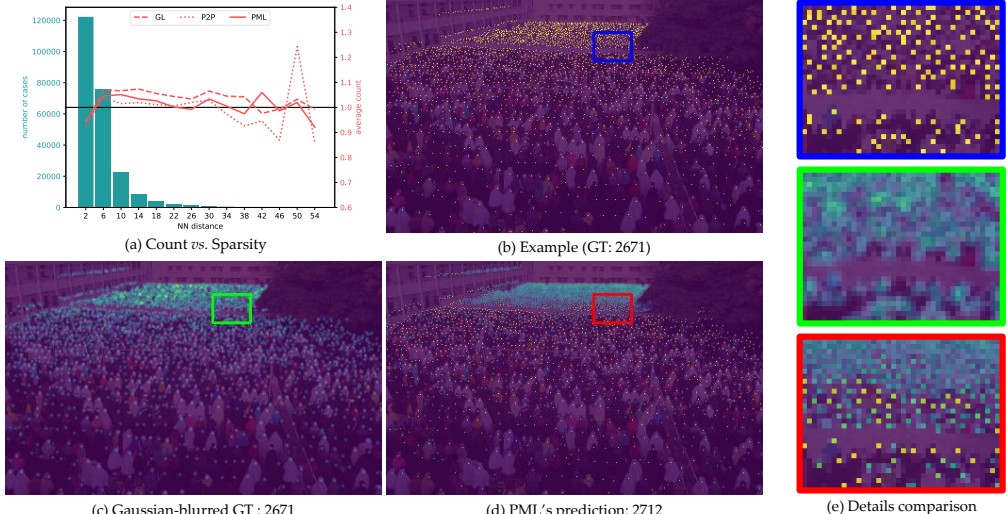

(a) Count *vs.* Sparsity

(b) Example (GT: 2671)

(c) Gaussian-blurred GT : 2671

(d) PML's prediction: 2712

(e) Details comparison

Figure 7: The performance difference of PML when handling sparse and dense crowds.

distance is very small (0∼4). the average count of PML (0.9441) is close to 1, and it is comparable to GL (0.9400, with intersection hypothesis) and P2PNet (0.9248, point-based counting).

We also give an example in UCF-QNRF dataset that contains both sparse and dense crowd in Fig. 7(b)-(e). Fig. 7(b) and (c) present the point location and the corresponding Gaussian-blurred GT map, while Fig. 7(d) demonstrates the prediction of PML. In sparse crowd, PML can predict density map similar to point map. In dense region, the density map is similar to a Gaussian-blured density map. There is a rectangle region clearly demonstrating how the prediction changes from point-similar case to density-similar case. These evidences illustrate that PML based on NN is able to capture the density information in both sparse and dense scene.

# D    IMPROVED P2PNET (SONG ET AL., 2021)

In the main paper, we have proven that P2PNet (Song et al., 2021) is also a special case of PML. However, there is a discrepancy between the implemented P2PNet (Song et al., 2021) and the derived (15): the counting network is optimized via *binary cross-entropy (BCE)* instead of L1 norm, which is inconsistent with the principle of SGL. To maintain consistency, we can define $\mathcal{D}$ as the Bregman divergence associated with the binary negentropy function $\varphi(\boldsymbol{x}) = \boldsymbol{x}^\top \log \boldsymbol{x} + (1-\boldsymbol{x})^\top \log(1-\boldsymbol{x})$, and then $\mathcal{D}_\varphi(\boldsymbol{p}, \boldsymbol{a})$ is instantiated as:

$$\mathcal{D}_\varphi(\boldsymbol{p}, \boldsymbol{a}) = \varphi(\boldsymbol{p}) - \varphi(\boldsymbol{a}) - \nabla\varphi(\boldsymbol{y})^\top(\boldsymbol{x} - \boldsymbol{y}) \tag{25}$$

$$= \boldsymbol{p}^\top \log\left(\frac{\boldsymbol{p}}{\boldsymbol{a}}\right) + (1-\boldsymbol{p})^\top \log\left(\frac{1-\boldsymbol{p}}{1-\boldsymbol{a}}\right), \tag{26}$$

which is the sum of pixel-wise binary KL divergence. Thus the BCE loss is derived accordingly:

$$\mathcal{L}'_{p2p} = \min_{\boldsymbol{p}\in\xi_{p2p}} \boldsymbol{c}^\top\boldsymbol{p} + \tau\mathcal{D}_\varphi(\boldsymbol{p}, \boldsymbol{a}) \tag{27}$$

$$= \min_{\boldsymbol{p}\in\xi_{p2p}} \boldsymbol{c}^\top\boldsymbol{p} - \tau \underbrace{\left[\boldsymbol{p}^\top \log\boldsymbol{a} + (1-\boldsymbol{p})\log(1-\boldsymbol{a})\right]}_{\text{binary cross entropy}} + P_c \tag{28}$$

where $P_c = \tau\left[\boldsymbol{p}^\top \log\boldsymbol{p} + (1-\boldsymbol{p})^\top \log(1-\boldsymbol{p})\right]$ is a constant.

From (28), we know how the BCE is derived from (5) by instantiating $\mathcal{D}$ with the cumulative binary KL divergence. Based on (28), $\boldsymbol{p}$ should also be re-derived. Specifically, $\boldsymbol{a}$ is modulated by a `Sigmoid`($\cdot$) in P2PNet (Song et al., 2021), i.e., $\boldsymbol{a} = [1 + \exp(-\boldsymbol{x})]^{-1}$. Therefore, (28) could be

| | MAE | MSE |
|---|---|---|
| P2PNet (Song et al., 2021) | 52.74 | 85.06 |
| P2PNet[†] | 52.49 | **83.02** |
| PML | **52.25** | 83.93 |

P2PNet[†] uses (30) to estimate $\boldsymbol{p}^*$.

Table 5: Test results on ShTech-A (Zhang et al., 2016) using different P2PNet.

| | MAE | MSE |
|---|---|---|
| DMC (Wang et al., 2020a) | 85.60 | 148.30 |
| $\tau = 1$ in (35) | 85.56 | 148.90 |
| $\tau = 8$ in (35) | 84.77 | 144.36 |
| PML (23) | **82.35** | **142.19** |

Table 6: Experimental results on UCF-QNRF (Idrees et al., 2018) using different DMC.

further expanded to:

$$\mathcal{L}'_{p2p} = \min_{\boldsymbol{p} \in \xi_{p2p}} \boldsymbol{c}^\top \boldsymbol{p} - \tau \boldsymbol{p}^\top \log\left(\frac{\boldsymbol{a}}{1-\boldsymbol{a}}\right) - \tau \mathbf{1}_{\hat{n}}^\top \log(1-\boldsymbol{a}) + P \tag{29}$$

$$= \min_{\boldsymbol{p} \in \xi_{p2p}} \underbrace{(\boldsymbol{c} - \tau \boldsymbol{x})^\top \boldsymbol{p}}_{\text{matching strategy}} - \underbrace{\tau \mathbf{1}_{\hat{n}}^\top \log(1-\boldsymbol{a}) + P}_{\text{constant}}, \tag{30}$$

whose solution is $\boldsymbol{p}^* = \delta(\arg\min_j c_j - \tau x_j)$. Compared with (17), $\mathcal{L}'_{p2p}$ in (30) prefers to use the prediction before the activation function instead of that processed by $\texttt{Sigmoid}(\cdot)$ to construct the cost matrix in the Hungarian algorithm.

In Tab. 5, we conduct experiments to compare how these two versions of P2PNet perform. P2PNet[†] is the one using (30) as the matching strategy to estimate $\boldsymbol{p}^*$. The results show that P2PNet[†] can further reduce the MAE to 52.49 from 52.74, and the MSE is also decreased by approximately 2.04. We also train the same model with PML. The third row of Tab. 5 demonstrates that PML can decrease the MAE to 52.25, while the MSE is also better than the vanilla P2PNet (83.93 *vs.* 85.06).

## E DISCUSSION ON DMC (WANG ET AL., 2020A)

In Sec. 4.4, we have shown how to implement PML using KL divergence and presented the relationship between PML and DMC (Wang et al., 2020a) in (21)∼(23). However, (23) is obtained by letting $\tau$ approach infinity. Here, we show the solution when $\tau$ is a positive real number by substituting (22) into (21):

$$\mathcal{L}_{dmc} = \boldsymbol{c}^\top \boldsymbol{p}^* + \tau \left( \boldsymbol{p}^{*\top} \log \frac{\boldsymbol{p}^*}{\boldsymbol{a}} - \boldsymbol{p}^{*\top} \mathbf{1} + \boldsymbol{a}^\top \mathbf{1} \right) \tag{31}$$

$$= \boldsymbol{c}^\top \boldsymbol{p}^* + \tau \left( \boldsymbol{p}^{*\top} \log \frac{\exp(-\boldsymbol{c}/\tau)}{\eta} + \boldsymbol{a}^\top \mathbf{1} - \boldsymbol{p}^{*\top} \mathbf{1} \right) \tag{32}$$

$$= \boldsymbol{c}^\top \boldsymbol{p}^* + \tau \left( -\frac{1}{\tau} \boldsymbol{p}^{*\top} \boldsymbol{c} - \log \eta + \boldsymbol{a}^\top \mathbf{1} - 1 \right) \tag{33}$$

$$= \tau \left( \boldsymbol{a}^\top \mathbf{1} - \log \eta - 1 \right) \tag{34}$$

Accordingly, the PML with a real postive number $\tau$ is formulated as:

$$\mathcal{L}_{dmc} = \boldsymbol{a}^\top \mathbf{1} - \log \eta, \quad \text{where} \quad \eta = \boldsymbol{a}^\top \exp(-\boldsymbol{c}/\tau). \tag{35}$$

Tab. 6 presents the experimental results of different DMCs. It shows that the performance of (35) is close to the vanilla DMC, but there is still a small gap. Setting $\tau = 8$ achieves lower estimation errors than the vanilla DMC. Besides, if we let $\tau \to \infty$ when computing $\boldsymbol{p}^*$, but use a constant $\tau = 2$ for the counting loss, *i.e.*, (23), the counting error will decrease significantly and be better than the vanilla DMC.

## F DIFFERENCE BETWEEN PML AND BL (MA ET AL., 2019)

In the main paper, we show that the difference between BL (Ma et al., 2019) and PML lies in the method used to compute the weight item in the *background loss*. In BL, $\boldsymbol{q}$ in 11 is constructed using only positive numbers, which considers all pixels to belong to the background and reduces the corresponding density values. In contrast, $\boldsymbol{c} - \bar{c}$ considers pixels meeting $c_i < \bar{c}$ as foreground and

| background loss | | MAE | MSE |
|---|---|---|---|
| BL (Ma et al., 2019) | $\boldsymbol{q}^\top \boldsymbol{a}$ | 88.7 | 154.8 |
| PML (variant) | $\boldsymbol{c}^\top \boldsymbol{a}$ | 85.3 | 146.1 |
| PML (ours) | $(\boldsymbol{c} - \bar{c})^\top \boldsymbol{a}$ | 76.6 | 132.2 |

Table 7: Test results on UCF-QNRF (Idrees et al., 2018) when using different background loss.

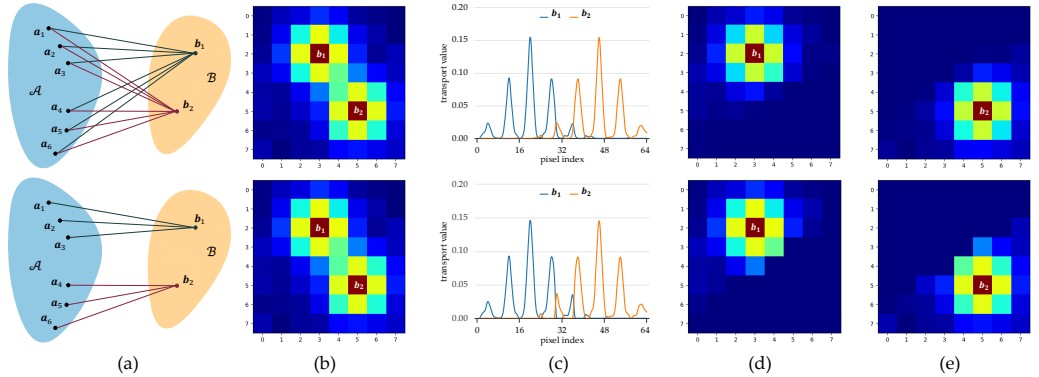

(a)  (b)  (c)  (d)  (e)

Figure 8: The difference between vanilla GL(Wan et al., 2021) (top) and the proposed PML (bottom). (a) $\mathcal{A}$ and $\mathcal{B}$ represent the predicted density map and GT point map. GL employs unbalanced optimal transport for dense assignment, but PML directly assigns the corresponding nearest neighbor pixels in $\mathcal{A}$ to each pixel in $\mathcal{B}$. (b) An example density map with two GT points $\boldsymbol{b}_1$ and $\boldsymbol{b}_2$. (c) Assigned density values in each pixel for $\boldsymbol{b}_1$ and $\boldsymbol{b}_2$. (d & e) The assigned pixel values or corresponding neighbors for $\boldsymbol{b}_1$ and $\boldsymbol{b}_2$. From (c-e) we find that the transport in UOT is very close to nearest neighbor assignment.

those with $c_i > \bar{c}$ as background. The foreground pixels are assigned negative weights to increase the corresponding density values, while positive weights are assigned to the background to lower the corresponding density values.

Here, we empirically demonstrate that the strategy in PML is superior to BL. In Tab. 7, we test two PML models and compare them with BL. The one without $-\bar{c}$ (the second row in Tab. 7) is similar to BL since it also considers all pixels as background and assigns positive weights to all pixels in the background loss part. Its performance is also close to BL. However, PML using $(\boldsymbol{c} - \bar{c})^\top \boldsymbol{a}$ as the background loss can significantly reduce the MAE, demonstrating its effectiveness. Besides, $\boldsymbol{c} - \bar{c}$ in PML has a solid theoretical basis since it is derived from proximal mapping.

## G CONNECTIONS & DIFFERENCE BETWEEN PML AND GL (WAN ET AL., 2021)

The generalized loss (GL) (Wan et al., 2021) is formulated as:

$$\mathcal{L}_{\mathbf{C}}^\varepsilon(\mathcal{A}, \mathcal{B}) = \min_{\mathbf{P} \in \mathbb{R}_+^{n \times m}} \underbrace{\langle \mathbf{C}, \mathbf{P} \rangle - \varepsilon H(\mathbf{P})}_{\text{entropic OT}} + \underbrace{\tau_1 \|\mathbf{P}\mathbf{1} - \boldsymbol{a}\|_2^2 + \tau_2 \|\mathbf{P}^\top \mathbf{1} - \mathbf{1}\|_1}_{\text{marginal penalties}}, \tag{36}$$

where $\mathbf{C}_{ij}$ measures the cost associated with moving a unit mass from $\boldsymbol{x}_i$ to $\boldsymbol{y}_j$ and is normally set as $\mathbf{C}_{ij} = \|\boldsymbol{x}_i - \boldsymbol{y}_j\|_2$. The hyperparameter $\varepsilon$ is a near-zero weight for the entropic term $H(\mathbf{P}) = -\sum_{ij} \mathbf{P}_{ij} \log \mathbf{P}_{ij}$. Additionally, two constants $\tau_1$ and $\tau_2$ are introduced to modulate the impact of these penalties for marginal deviation. Once the optimal transport plan $\widehat{\mathbf{P}}$ is obtained, the gradient $\frac{\partial \mathcal{L}}{\partial \boldsymbol{a}}$ is accessible for backpropagation. However, there are no applicable algorithms to solve (36). The alternative is to implement an unbalanced OT with KL divergence as the marginal penalties to approximate $\widehat{\mathbf{P}}$ via Sinkhorn algorithm (Peyré et al., 2019), and then substitute it into (36) to obtain $\mathcal{L}_{\mathbf{C}}^\varepsilon$ for automatic differentiation.

*Connections.* Fig. 8, we visualize how UOT and nearest neighbor perform the assignment in loss computation. Fig. 8(a) illustrates the transport plan in PML is defined by assigning each density

Figure 9: The structure of the trained model in our experiments.

pixel to its nearest GT point. Consequently, PML only needs to compute the loss within each point-neighbor group separately, *i.e.*, within just one point in GT and its neighbors in the density map. In this case, the Sinkhorn algorithm(Peyré et al., 2019) is not required, as the mass for each pixel is transported completely to the same target point.

Comparing UOT with nearest neighbor, it is easily observed that UOT still obeys the intersection hypothesis, since it "softly" assigns one pixels to multiple points in GT. However, nearest neighbor only assigns each density pixels to its nearest point, which could be considered as a "hard" assignment. The nearest neighbor can be considered as the solution of (36) when $\varepsilon = 0$, $\tau_1 = \infty$ and $\tau_2 = 0$. This means the nearest neighbor sacrifices the second marginal penalty, *i.e.*, the count loss $\|\mathbf{P}^\top \mathbf{1} - \mathbf{1}\|_1$, to approximate $\hat{P}$. However, PML (24) implicates the count loss, mending the flaw.

*Drawbacks of GL.* We find that GL has two drawbacks. Firstly, the theoretical analysis of UOT that relates it to L2 and BL is imperfect. The proof stems from applying a half Sinkhorn iteration to obtain an approximate UOT solution, which means that the equivalence of UOT to L2 and BL is based on computing a plan non-iteratively, using a single pass over the inputs. Thus the connection is through a static construction of an approximate $\widehat{\mathbf{P}}$, instead of the whole Sinkhorn iterations, *i.e.*, the full UOT solution. Secondly, the Sinkhorn algorithm in implementation is time-consuming and requires a large amount of space to store the transport plan during computation, limiting training efficiency.

*Efficiency of PML.* Theoretically, the computation of UOT is $\mathcal{O}(tmn)$, where $m \times n$ is the shape of the cost matrix, and $t$ is the number of iterations of the Sinkhorn algorithm. Meanwhile, the nearest neighbor only requires $\mathcal{O}(mn)$ to get the index of the nearest point in GT for each pixel in the prediction. In practice, the *average* loss computation time ratio is $\frac{\text{GL}}{\text{PML}} = \frac{0.045s}{0.013s} \approx 3.46$ on a 3090Ti GPU, with the density map resolution of $256 \times 256$.

## H MODEL STRUCTURE

Fig. 9 displays the model structure of our counter. VGG19 (Simonyan & Zisserman, 2015) or HRNet (Wang et al., 2020b) are adopted as the backbone to extract image features. After that, feature maps outputted from different blocks are fused to generate the density features. Based on this, several convolutional layers are stacked to predict the final density map, and pixel shuffle (Shi et al., 2016) is applied to interpolate the final prediction to a resolution that is half of the input. For VGG19, we fuse features from the last two blocks. For HRNet, we fuse features maps from the first two branches.

## I ABLATION STUDIES

*The effect of $\epsilon$.* In (24), a balanced hyperparameter $\epsilon$ is introduced to weigh the two components, enabling a focus on either localization or counting. Fig. 10(a) shows the impact of $\epsilon$ on the counting performance of the trained model. A value of $\epsilon = 1$ yields a good counting model, but surprisingly, $\epsilon = 2$ results in even lower estimation errors. However, further increasing this hyperparameter proves detrimental to training, as the model leans excessively towards counting, neglecting vital localization information: setting $\epsilon$ to 4 and 8 yields a degradation in counting performance compared to $\epsilon = 1$.

*The effect of upsampling module.* We conducted an ablation study on the upsampling modules, namely bilinear interpolation and pixel shuffle (Shi et al., 2016), to systematically assess their impact on model performance. Bilinear interpolation is a non-learnable method, while pixel shuffle is a

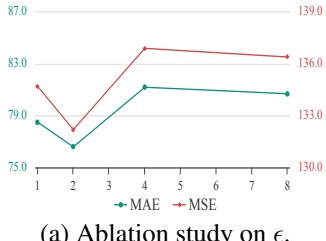
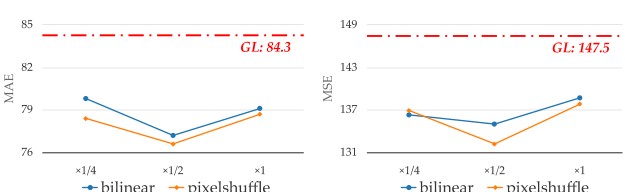

(a) Ablation study on $\epsilon$.  (b) Ablation study on upsampling module.

Figure 10: The effect of $\epsilon$ in (24) and upsampling module.

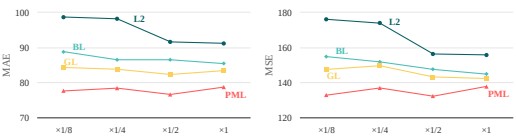

Figure 11: The relationship between losses and output resolution.

Table 8: Results on additional models with differents losses.

| METHOD | | ShTech A | | ShTech B | | JHU | |
|---|---|---|---|---|---|---|---|
| | | MAE | MSE | MAE | MSE | MAE | MSE |
| MCNN | BL | 103.0 | 168.6 | 16.3 | 28.2 | 105.2 | 358.19 |
| | GL | 100.4 | 165.7 | 16.0 | 26.7 | 99.5 | 339.6 |
| | PML | 95.9 | 161.5 | 15.2 | 24.6 | 95.9 | 314.2 |
| CSRNet | BL | 65.8 | 115.0 | 8.9 | 13.5 | 72.9 | 287.5 |
| | GL | 64.8 | 111.9 | 8.1 | 12.2 | 69.7 | 275.3 |
| | PML | 64.1 | 109.3 | 7.7 | 11.6 | 66.5 | 268.8 |

learnable module often utilized in single-image super-resolution tasks. The detailed structure is provided in the Supplementary. The test results in Fig. 10(b) show how the output with different interpolation rates ($\times \frac{1}{4}$, $\times \frac{1}{2}$, and $\times 1$) affects counting performance. Here, $\times r$ denotes that the output resolution is $r$ times that of the input image. Due to its learnable nature, pixel shuffle consistently achieves lower errors in all experiments when compared to bilinear interpolation. Although the differences in estimation errors between different resolutions are marginal, the model with an output resolution rate of $\frac{1}{2}$ has the optimal performance during training.

Fig. 11 presents the results when different losses are applied to different output sizes. Overall, a detailed density map is beneficial for counting model since it provides a more detailed prediction. However, Fig. 11 shows that the output size does not affect the superiority of PML.

*More comparison on loss functions.* In Tab. 8, we compare MCNN (Zhang et al., 2016) and CSR-Net (Li et al., 2018) on more datasets than the main paper. Without loss of generalization, PML still outperforms BL (Ma et al., 2019) and GL (Wan et al., 2021) when other networks are adopted.

## J DETAILS ON NWPU BENCHMARK

Tab. 9 presents the details on the NWPU benchmark to understand in which parts PML works better than other methods. In the counting section (the top part of Tab. 9), the overall performance of PML is close to STEERER. However, both models excel at different scene levels. PML performs the best in S3, whose crowd size variance is the largest (GT count $\in (500, 5000]$), and achieves similar performance to STEERER in other scenes. In the benchmark of localization, FIDTM achieves superior performance in A0 (head area $< 10$) and A1 (head area $\in (10, 100]$), typical dense scenes. However, the Recalls of PML in A0 and A1 are the second best ones, and PML obtains the best recall from A2 to A4, where the head area ranges from 100 to $10^5$. In the extremely sparse scene A5 (head area $> 10^5$), the segmentation-based method, TopoCount (Abousamra et al., 2021), achieves the best results. These findings indicate that different models excel in different scenarios, but PML demonstrates strong performance in the localization task.

This section also presents some visualization results are presented in Fig. 12 to Fig. 15.

| Method | (backbone) | overall | | Scene Level (MAE / MSE) | | | | |
|---|---|---|---|---|---|---|---|---|
| | | mae | mse | S0 | S1 | S2 | S3 | S4 |
| FIDTM | (HRNet) | 86.0 | 312.5 | **21.6 / 129.2** | 13.70 / 20.03 | 55.6 / 174.9 | 217.1 / 391.9 | **1645.4 / 2288.2** |
| P2PNet | (vgg16bn) | 72.6 | 331.6 | 34.7 / 202.7 | 11.3 / 17.0 | 31.5 / 52.3 | 161.0 / 265.8 | 2311.6 / 2960.7 |
| GL | (vgg19) | 79.3 | 346.1 | 92.4 / 461.5 | 8.2 / 13.9 | 34.4 / 65.6 | 179.2 / 299.2 | 2228.3 / 2865.8 |
| STEERER | (HRNet) | **63.7** | 309.8 | 48.3 / 327.3 | **6.0 / 10.8** | **25.9 / 51.4** | 158.3 / 275.7 | 1814.5 / 2636.9 |
| PML | (vgg19) | 73.6 | 338.6 | 39.6 / 225.7 | 6.7 / 12.0 | 32.8 / 93.4 | 179.6 / 296.9 | 2141.7 / 2934.3 |
| PML | (HRNet) | 63.8 | **306.9** | 26.0 / 198.4 | 7.0 / 13.3 | 27.4 / 51.7 | 151.7 / 262.5 | 1983.0 / 2707.7 |

| Method | (backbone) | overall (×100) | | | Head Area (Rec. ×100) | | | | | |
|---|---|---|---|---|---|---|---|---|---|---|
| | | F1. | Prec. | Rec. | A0 | A1 | A2 | A3 | A4 | A5 |
| TopoCount | (vgg16) | 69.2 | 68.3 | 70.1 | 5.7 | 39.1 | 72.2 | 85.7 | 87.3 | **89.7** |
| FIDTM | (HRNet) | 75.5 | 79.8 | 71.7 | **22.8** | **66.8** | 76.0 | 71.9 | 37.4 | 10.2 |
| IIM | (HRNet) | 76.0 | **82.9** | 70.2 | 11.7 | 45.3 | 73.4 | 83.0 | 64.5 | 16.7 |
| GL | (vgg19) | 66.0 | 80.0 | 56.2 | 3.6 | 21.2 | 56.2 | 79.4 | 79.4 | 51.2 |
| STEERER | (HRNet) | 77.0 | 81.4 | 73.0 | 12.0 | 46.0 | 73.2 | 85.5 | 86.7 | 64.3 |
| PML | (vgg19) | 73.5 | 77.6 | 69.8 | 5.5 | 40.7 | 72.3 | 84.1 | 82.7 | 62.4 |
| PML | (HRNet) | **80.2** | 80.9 | **79.5** | 18.4 | 63.8 | **80.5** | **88.2** | **88.3** | 69.9 |

Table 9: Details on the NWPU counting (top) and localization (bottom) benchmark.

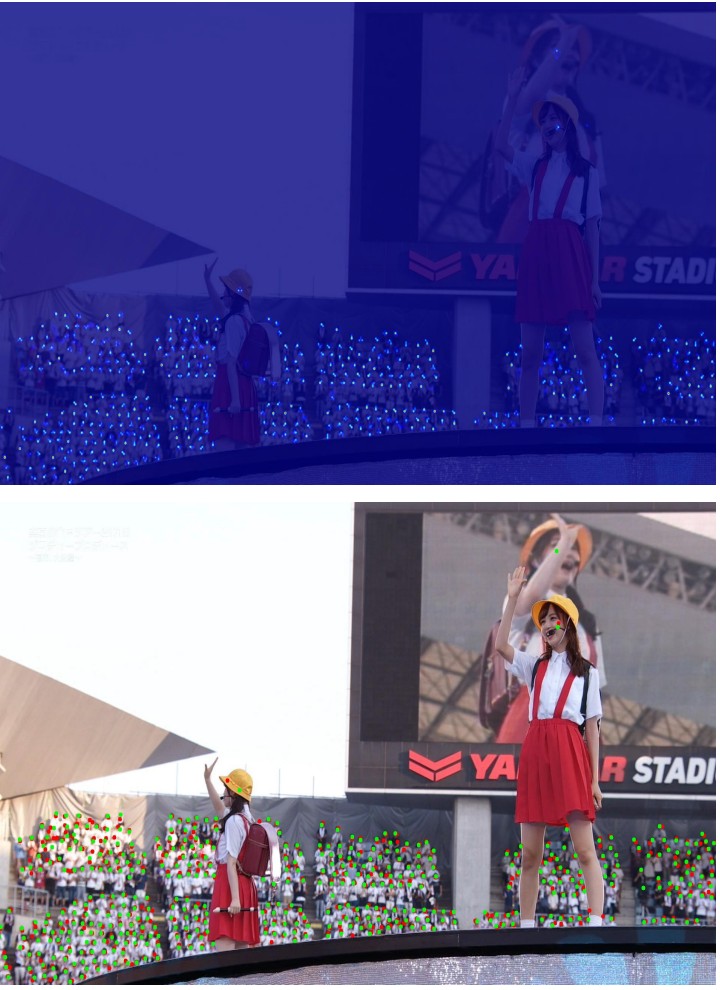

Figure 12: Density map and localization map visualization.

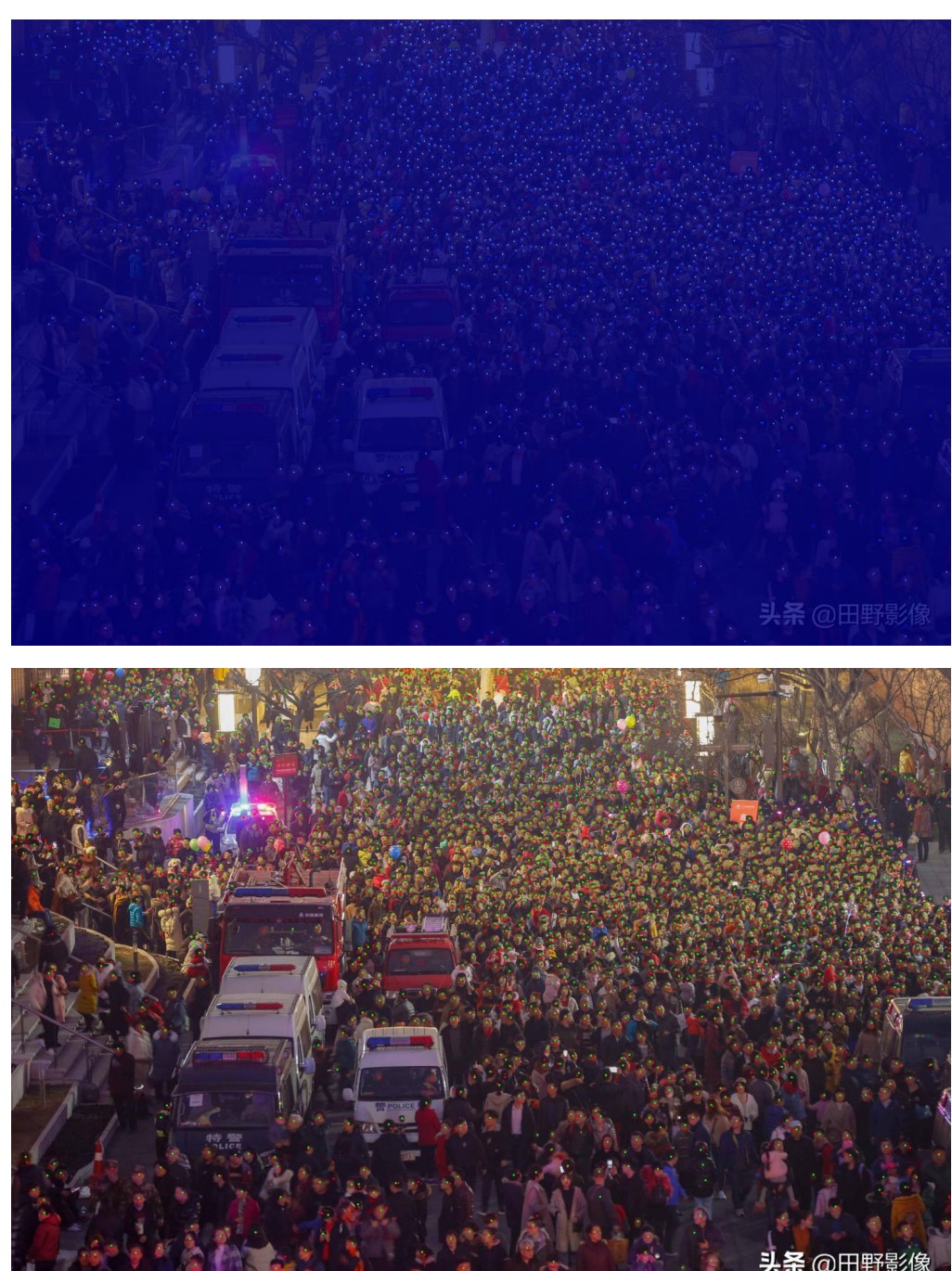

Figure 13: Density map and localuization map visualization.

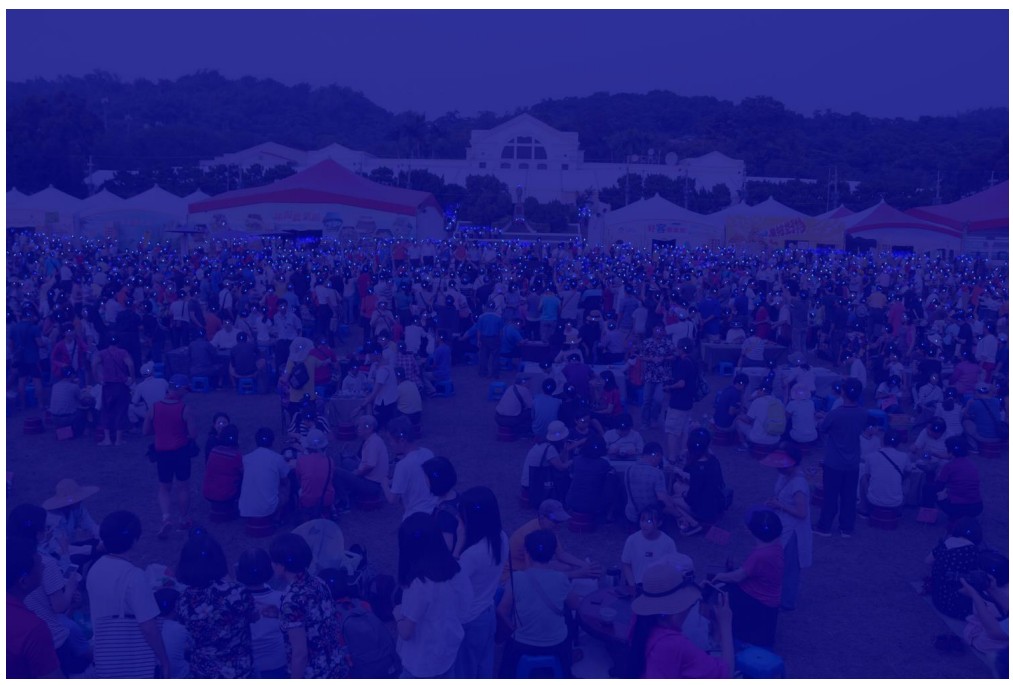

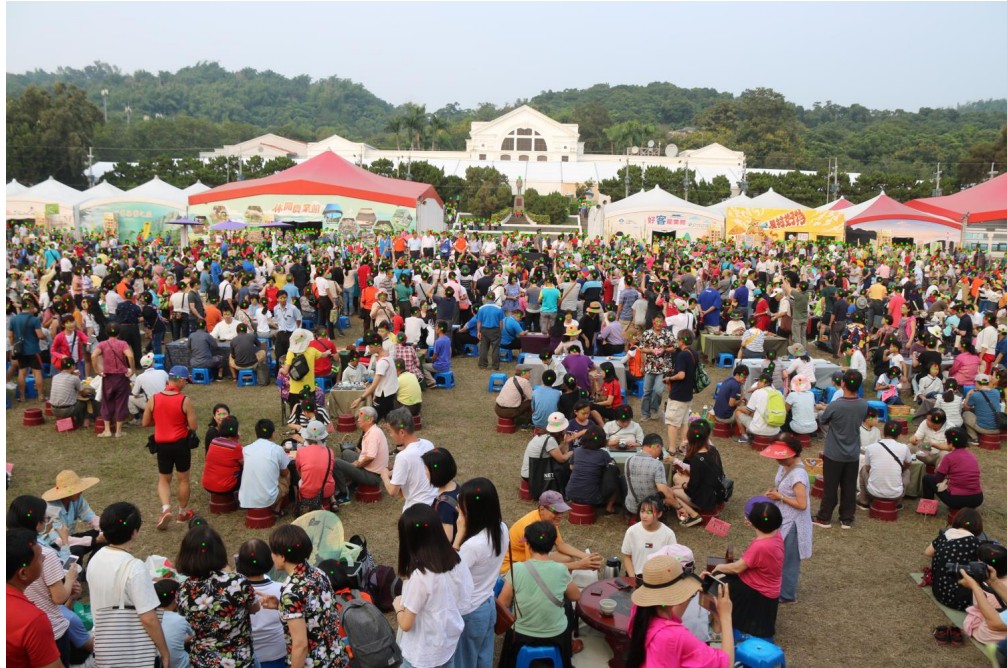

Figure 14: Density map and localuization map visualization.

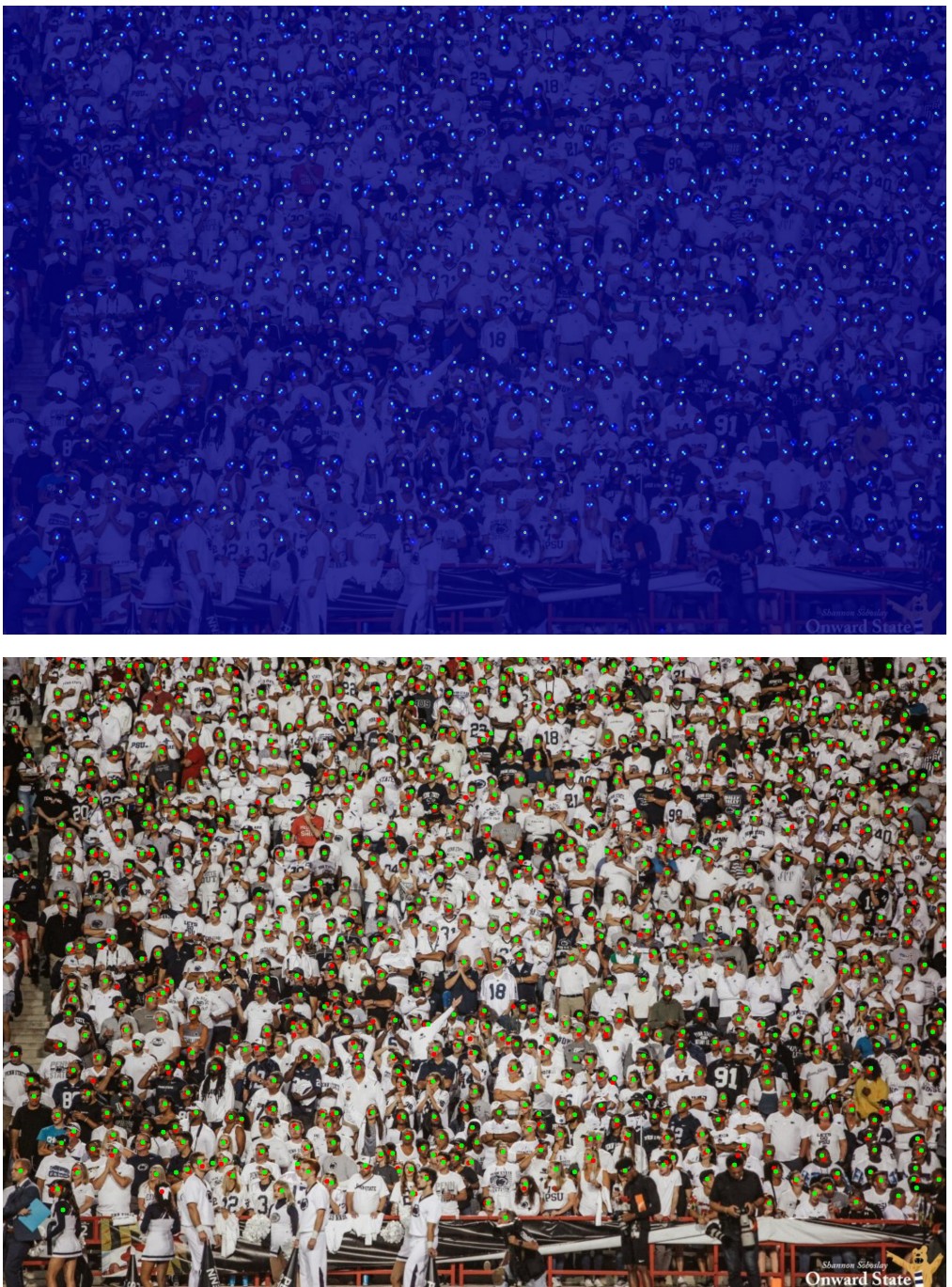

Figure 15: Density map and localuization map visualization.

