# OpenReview forum: "Proximal Mapping Loss: Understanding Loss Functions in Crowd Counting & Localization"
_ICLR.cc/2025/Conference — ICLR 2025 Poster_

### Official Review · Reviewer_fWss · 2024-10-27

**Soundness:** 3
**Presentation:** 3
**Contribution:** 2
**Rating:** 6
**Confidence:** 5

**Summary:**

This paper argues that the intersection hypothesis in existing density-map based counting methods is consistent with reality, where one pixel is influenced by multiple ground-truth points. To eliminate such a hypothesis, the authors propose a proximal mapping loss, which follows the philosophy of divide-and-conquer. Specifically, the predicted density map is first divided into non-overlapping irregular patches. Then, proximal mapping loss (PML) is computed within each point-neighbor case. Experimental results on several crowd counting datasets demonstrate the effectiveness of the proposed method. Additionally, this paper also shows that the proposed PML compasses many existing loss functions.

**Strengths:**

1. This paper presents an interesting argument about the intersection hypothesis in existing density-regression counting methods.
2. The paper proposes a divide-and-conquer strategy to eliminate the intersection hypothesis, featuring a proximal-mapping loss.
3. Experiments show that the proposed method achieves state-of-the-art counting and localization results.

**Weaknesses:**

1. The main concern is the training efficiency. The proposed method not only needs to compute the nearest neighbor, but also constructs learning targets for each sub-problem presented in Eq. 1. This could be computationally expensive when dealing with congested scenes.
2. The robustness to noisy annotations needs further clarifications, e.g., performance comparisons with existing methods would be sufficient.

**Questions:**

1. Does the proposed method suffer from slow convergence speed? As shown in Fig. 4, the training epoch reaches 2000.
2. Did the authors use different epsilon for different datasets? Fig. 8 shows that the choice of epsilon does affect the performance.

---

> ### Comment · Reviewer_fWss · 2024-11-22
>
> After reading the comments from fellow reviewers, I agree that additional comparisons and discussions are necessary to support the effectiveness of the proposed approach. Given that the authors did not provide a rebuttal, these concerns remain unaddressed. Therefore, I am inclined to recommend rejecting this paper.

---

> > ### Author Response · Authors · 2024-11-22
> > **Looking forward to your responses or further suggestions/comments!**
> >
> > Dear Reviewer fWss,
> >
> > Thanks for all of your suggestions and sorry for the late reply. We just posted [our responses to your question](https://openreview.net/forum?id=7p8CcxP1Xc&noteId=Q31uPm4DM6). Please don’t hesitate to let us know if there are any additional clarifications or evidence we can offer, as we would love to convince you of the merits of the paper. Thanks!
> >
> > -the authors

---

> > > ### Comment · Reviewer_fWss · 2024-11-25
> > >
> > > Thanks for the response. The rebuttal has addressed most of the concerns. Regarding the robustness to noisy annotations, a quantitative comparison with existing methods (e.g., Fig. 4 in NoiseCC) is preferred.

---

> > > > ### Author Response · Authors · 2024-11-26
> > > > **Response to Reviewer fWss -- part 2**
> > > >
> > > > Thanks for your suggestion. We revise the paper again and answer your newest comments here:
> > > >
> > > > **Q5: a quantitative comparison with existing methods (e.g., Fig. 4 in NoiseCC)**
> > > >
> > > > **A5:** The way to add noise in the last revised version follows that in BL, so we compare it with BL in the newest version (Fig. 4(a)). Besides, in the newest revised version, we replace Fig.4(b) with the comparison that adding noise follows NCC. NCC performs better than the PML with L2-norm, but PML with L1-norm achieves lower estimation errors.

---

> > > > > ### Comment · Reviewer_fWss · 2024-11-26
> > > > >
> > > > > Thanks for the response. Given that the rebuttal has addressed the concerns, I will retain my initial score.

---

> ### Author Response · Authors · 2024-11-22
> **Response to Reviewer fWss**
>
> **Q1:  training efficiency demonstration, especially the nearest neighbor and learning objective for each sub-problem.**
>
> **A1:** In (24), we present how to compute the PML via matrix operations, which is executed fast on GPU. The nearest neighbor can be easily computed by getting the column minimum of the cost matrix, while all sub-problems can be solved in parallel. All operations involved in PML have stable and fast implementation using GPU.
>
> In the last paragraph of Sec. G in the Appendix (line-945 to line-949), we compare the efficiency of PML to GL, and show that the computation ratio in practice is 3.46 on a 3090 Ti GPU. Besides, when training HRNet on the UCF-QNRF dataset, PML converges faster and achieves an MAE smaller than 75 in 1100 epochs, while GL requires more than 2170 epochs to achieve an MAE smaller than 80. For each epoch, GL needs around 90 seconds, but PML only requires around 40 seconds.
>
> ---
>
> **Q2:  The robustness to noisy annotations.**
>
> **A2:** The robustness of PML with L1-norm is claimed when compared with L2-norm. See *line-318* to *line-328*, where we explain why L1-norm performs better than L2-norm when used as the count loss. In the revised version, we add uniform noise to human annotations to test the robustness of the L1-norm when compared with the L2-norm. As shown in the updated Fig.4(a)-(b), L1-norm achieves lower estimation errors than L2-norm in all noise degrees.
>
> In our study, we claim that the reason for this phenomenon is that the L1-norm is more robust to label noise than the L2-norm because the former sets the learning objective according to the predicted count, which is consistent with the final evaluation metric. Fig. 5 provides a detailed visualization of the learning objective when PML uses L1-norm and L2-norm. Specifically, the comparison between Fig. 5(d) and Fig. 5(g) illustrates this point. When the count is close to 1 (well-estimated), the learning objective of L1-norm is close to the prediction (Fig. 5(g)), resulting in a small loss, which indicates that the training is good enough. However, the learning objective of L2-norm always forces the prediction to be close to the GT point's location, leading to a large loss even though the prediction is close to 1 (Fig. 5(d)). This latter case is harmful to the counting task because it forces the prediction to move to an inappropriate position even when the count is well-estimated.
>
> ---
>
> **Q3:  Does PML suffer from slow convergence speed?**
>
> **A3:** PML has a fast convergence speed. In the following table, we present after how many epochs the MAE of HRNet on the validation set is smaller than a specific value:
>
> |MAE | < 100 | < 90 | < 85 | < 80 | < 75 |
> | :---: | :-----: | :----: | :----: | :----: | :----: |
> | L2 | 1610 |  -  |  -  | -    | -    |
> | BL |  220  | 640 | 1880 | -    | -    |
> | DMC |  200  | 420 | 1120 | -    | -    |
> | GL |  130  | 280 | 970 | 2170 | -    |
> | PML |  40  |  80  | 200 | 410  | 1070 |
>
> Comparing PML to GL, GL achieves an MAE smaller than 85 after 970 epochs, while PML only requires 200 epochs to reach it and achieves an MAE smaller than 75 after 1070 epochs. Besides, GL needs around 90 seconds for each epoch, but PML only requires around 40 seconds. These pieces of empirical evidence demonstrate that PML converges fast.
>
> ---
>
> **Q4:  Did the authors use different $\epsilon$ for different datasets?**
>
> **A4:** No, we use the same $\epsilon = 2$ for all experiments since the ablation study (Fig. 10(a)) shows that the best performance is achieved when it is set to 2 on the UCF-QNRF dataset.

---

### Official Review · Reviewer_opsr · 2024-11-01

**Soundness:** 3
**Presentation:** 3
**Contribution:** 2
**Rating:** 6
**Confidence:** 4

**Summary:**

This paper proposes a proximal loss function for crowd counting and localization. The key idea is to divide the predicted density map into multiple non-overlapped patches and compute loss for each patch via proximal mapping. Such an operation eliminates the intersection hypothesis in density regression-based methods, which improves the robustness and accuracy of the counting models. Evaluations on several crowd counting datasets demonstrate the superiority of the proposed loss over previous loss functions.

**Strengths:**

* This paper presents a new proximal mapping loss, eliminating the intersection hypothesis in existing density-map based methods.
* The authors comprehensively discuss the relationship between the proposed loss and existing loss functions.
* Extensive experiments validate the effectiveness of the proposed loss.

**Weaknesses:**

* It would be better to give a quantitative comparison of training efficiency between the proposed loss and existing losses. While the authors mention that the loss computation of PML is more efficient than GL, it is unclear whether the convergence speed of PML is faster than existing loss functions.
* It appears that retaining the spatial resolution is important to make PML work as occluded objects could degenerate into a single point when downsampling. In this case, it is beneficial to analyze the characteristics of the proposed loss under different crowd densities.
* Following the previous comments, it is suggested to discuss in what scenarios PML is better than previous methods.

**Questions:**

* Did the authors use the same output resolution to train different models in Table 4? As shown in the paper, spatial resolution could affect the performance.
* Does PML converge faster than existing methods? This is important when applying the proposed method to large-scale datasets.

---

> ### Author Response · Authors · 2024-11-22
> **Response to Reviewer opsr**
>
> **Q1: a quantitative comparison of training efficiency**
>
> **A1:** The efficiency is compared in the last paragraph of Sec. G in the Appendix, from line 944 to line 949. Overall, the computation ratio in practice is 3.46 on a 3090 Ti GPU. Besides, when training HRNet on the UCF-QNRF dataset, PML converges faster and achieves an MAE smaller than 75 in 1100 epochs, while GL requires more than 2100 epochs to achieve an MAE smaller than 80. For one epoch, GL needs around 90 seconds, but PML only requires around 40 seconds.
>
> ---
>
> **Q2:  analyze the characteristics of the proposed loss under different crowd densities.**
>
> **A2:** Spatial resolution affects not only PML but also other methods. Fig. 11 presents "the estimation errors" vs. "the resolution" for all loss functions. It shows that larger resolution results in better performance. To show the performance of PML under different crowd densities, we also added the detailed performance on the NWPU counting and localization benchmarks in Sec. J and Tab. 9 in the revised appendix. PML achieves the best performance in S3, where the count ranges from 500 to 5000. In the localization benchmark, PML achieves the best or second-best performance on all levels.
>
> ---
>
> **Q3: Which scene PML works the best?**
>
> **A3:** As displayed in Tab. 9 in the Appendix, PML achieves the best counting performance in S3, where the counting range is from 500 to 5000 in the NWPU counting benchmark. In the localization benchmark, PML achieves the best or second-best performance on all levels.
>
> Besides, we added a new section (Sec. C) and a new figure (Fig. 7) to show how PML processes dense and sparse crowds. As displayed in Fig. 7(a), these lines demonstrate the "average count in each point-neighbor case" vs. the "NN distance of each point." PML's average count is closer to 1 when the distance between the concerned point and its nearest point ranges from 20 to 28.
>
> ---
>
> **Q4:  Is the output resolution the same to train different models in Tab. 4?**
>
> **A4:**  The output resolutions are the same for MCNN, CSRNet, and HRNet, as we use the same structure as that in the original papers, and HRNet's structure is not changed. For VGG19, we borrow the results from the original paper in other loss function studies (resolution "X1/8" in Fig. 11). The initialized version of PML used the resolution of "X1/2" in Tab. 4.
>
> Thanks for the kind reminder. We have revised the results and now present the "X1/8" results in Tab. 4 in the new version (*line-437*). However, the conclusion with the "X1/8" map is consistent with the previously presented results: PML surpasses other loss functions.
>
> ---
>
> **Q5: Does PML converge faster than existing methods?**
>
> **A5:** PML has a fast convergence speed. In the following table, we present after how many epochs the MAE of HRNet on the validation set is smaller than a specific value:
>
> |MAE | < 100 | < 90 | < 85 | < 80 | < 75 |
> | :---: | :-----: | :----: | :----: | :----: | :----: |
> | L2 | 1610 |  -  |  -  | -    | -    |
> | BL |  220  | 640 | 1880 | -    | -    |
> | DMC |  200  | 420 | 1120 | -    | -    |
> | GL |  130  | 280 | 970 | 2170 | -    |
> | PML |  40  |  80  | 200 | 410  | 1070 |
>
> Comparing PML to GL, GL achieves an MAE smaller than 85 after 970 epochs, while PML only requires 200 epochs to reach it and achieves an MAE smaller than 75 after 1070 epochs. Besides, GL needs around 90 seconds for each epoch, but PML only requires around 40 seconds. These pieces of empirical evidence demonstrate that PML converges fast.

---

> > ### Comment · Reviewer_opsr · 2024-11-25
> >
> > Thanks for the rebuttal. My concerns have been well addressed. I would like to keep my initial rating.

---

### Official Review · Reviewer_cGtL · 2024-11-02

**Soundness:** 3
**Presentation:** 3
**Contribution:** 2
**Rating:** 6
**Confidence:** 5

**Summary:**

This paper proposes the Proximal Mapping Loss (PML), a novel density regression loss for crowd counting and localization. PML studies the so-called intersection hypothesis of classic density map and addresses the intersection issue by dividing the predicted density map into multiple patches through nearest neighbor analysis. It then constructs a dynamic learning target for each case, leading to more tractable and robust training. In addition, the paper also shows the theoretical connection between the PML loss and other closely related loss functions. Experiments were conducted to demonstrate the efficacy of PML in both crowd counting and localization.

**Strengths:**

+ **Good motivation**. The paper targets an appropriate problem in crowd localization, featured by the intersection hypothesis. The problem sounds.
+ **A novel loss**. To evade the sum property of density map, a new loss function is proposed to constrain the regression target by dividing the training sample into many point-neighbor cases and then computing sub-loss by comparing the difference between the prediction and a dynamic learning target defined via proximal mapping. According to Fig. 2, it does advance the prior Bayesian Loss and Generalized Loss in discriminating close individuals.
+ **Theoretically sound**. The derivations in Sec. 3.2 show how the loss function works in a relatively intuitive way during training, and its connections with several loss functions and training schemes have been proven, which is theoretically sound.
+ **SOTA performance**. Through evaluations on several benchmark datasets, PML is shown to outperform current state-of-the-art methods in general.

**Weaknesses:**

- **Some closely related work is missing**. For example, the same problem of intersection hypothesis has been discussed in [a]. While the solution differs, the way for addressing such an issue in [a] could be discussed and compared in Fig.1 as well.

[a] Decoupled two-stage crowd counting and beyond, IEEE Transactions on Image Processing, 2021.

- **Inappropriate baselines**. Since the main focus of the work is the loss function, it should demonstrate its effectiveness on SOTA models instead of constructing a new one (the HRNet based model). It would be difficult to interpret where the real improvement comes from.
- Some important experiments are missing such that some results are not directly comparable. For example, the proposed approach should report the results of vgg-16bn as well.

- **A proof-of-concept experiment is missing**. The initial claim of the paper is to solve the intersection hypothesis problem, but it turns out to focus on how to better predict points. There lacks some comparative experiments to prove the soundness of the initial claim.
- The paper claims that PML is robust to label noise. Though visualization results are given to demonstrate its robustness, there lacks evidence to support this.

Minor issues：
- Some cited results may be wrong. E.g., CSRNet in Table 2 uses vgg-16 rather than vgg-16bn.

**Questions:**

- Please consider rephasing the intersection hypothesis. It does not make sense to me that in reality one pixel would not contain two objects. If two objects occlude, one pixel can certainly contain part of the two objects.
- Further ablation experiments should be conducted to justify the effect of noise on the model and the effectiveness of PML in coping with it.
- The paper has demonstrated its work on removing the intersection hypothesis. Experiments, however, has shown nothing related to the hypothesis and mainly focus on the efficacy of PML. Could you explain through experiments or visualizations whether PML really addresses such an issue?

- Table 4 only shows plug-and-play results for MCNN and CSRNet; they are somewhat outdated. It is better to report the use of PML on recent SOTA models.

- It would be good to showcase how much PML can actually eliminate the intersection hypothesis? Can it be shown in a quantitative way?

**Details Of Ethics Concerns:**

N.A.

---

> ### Author Response · Authors · 2024-11-22
> **Response to Reviewer cGtL -- part 1**
>
> **Q1:  Miss discussion with D2CNet[a]: Decoupled two-stage crowd counting and beyond, IEEE Transactions on Image Processing, 2021.**
>
> **A1:** Thanks for the reminder. We have added and discussed D2CNet in the Introduction (*line 86*) and Related Work (*line 153*). Additionally, FIDTM and iKNN are also discussed, as they were initially overlooked (*lines 77 and 167*).
>
> ---
>
> **Q2: Demonstrate its effectiveness on SOTA models instead of constructing a new one (the HRNet-based model)**
>
> **A2:** HRNet is indeed the backbone of the current SOTA -- STEERER. STEERER uses HRNet as the backbone but incorporates a feature selection and inheritance module and a specific training scheme called masked selection and inheritance loss. These elaborate designs aim to fuse density maps with multiple resolutions.
>
> | Model | MAE  | MSE  |
> | :---------------: | :--: | :--: |
> |     STEERER (#1)  | 58.4 | 92.0 |
> |     STEERER (#2)  | 56.5 | 92.8 |
> |     STEERER (#4)  | 54.6 | 86.9 |
> |       PML (#1)    | **52.3** | **84.7** |
>
> In the table, "(#n)" means that there are n density maps from different resolutions fused in STEERER via feature selection. "(#1)" means that feature selection is not applied.
>
> The comparison shows that PML, which outputs just one density map without any complicated feature selection module, achieves better performance than STEERER, which fuses 4 density maps with various resolutions using elaborate neural network structures.
>
> ---
>
> **Q3:  vgg16bn results.**
>
> **A3:** Thanks for the comment. We have added it to Tab.2 in the revised version.
>
> ---
>
> **Q4:  This paper aims to solve the intersection hypothesis problem, but the paper lacks experiments to prove it.**
>
> **A4:** The intersection hypothesis is not a problem, but a design principle. In the Introduction, we claimed that most crowd counting methods are trained following the framework of density map regression, whose learning objective is also a density map where each foreground pixel in it is shared by multiple points. We summarized this as the intersection hypothesis. In the Related Work section, we introduce previous methods that are learned with (line 131) and without (line 157) the intersection hypothesis. Our PML follows the density map regression framework but is trained without the intersection hypothesis. To demonstrate how the intersection hypothesis is removed in the divide stage of PML, we added a new figure (Fig. 6) to present how the density map is split into many irregular patches.
>
> ---
>
> **Q5:  PML is robust to label noise.**
>
> **A5:** The robustness of PML with L1-norm is claimed when compared with L2-norm. See *line-318* to *line-328*, where we explain why L1-norm performs better than L2-norm when used as the count loss. In the revised version, we add uniform noise to human annotations to test the robustness of the L1-norm when compared with the L2-norm. As shown in the updated Fig.4(a)-(b), L1-norm achieves lower estimation errors than L2-norm in all noise degrees.
>
> In our study, we claim that the reason for this phenomenon is that the L1-norm is more robust to label noise than the L2-norm because the former sets the learning objective according to the predicted count, which is consistent with the final evaluation metric. Fig. 5 provides a detailed visualization of the learning objective when PML uses L1-norm and L2-norm. Specifically, the comparison between Fig. 5(d) and Fig. 5(g) illustrates this point. When the count is close to 1 (well-estimated), the learning objective of L1-norm is close to the prediction (Fig. 5(g)), resulting in a small loss, which indicates that the training is good enough. However, the learning objective of L2-norm always forces the prediction to be close to the GT point's location, leading to a large loss even though the prediction is close to 1 (Fig. 5(d)). This latter case is harmful to the counting task because it forces the prediction to move to an inappropriate position even when the count is well-estimated.
>
> ---
>
> **Q6:  CSRNet use vgg16 not vgg16-bn.**
>
> **A6:**  Thanks for the kind reminder, we have revised it in the new version, see *line-380*.
>
> ---

---

> ### Author Response · Authors · 2024-11-22
> **Response to Reviewer cGtL -- part 2**
>
> **Q7: rephrasing the intersection hypothesis.**
>
> **A7:** Due to the view perspective, one pixel may contain parts of two objects in the given image. However, the role of the intersection hypothesis is **to define the learning objective, not to model the image properties.** The learning objective is a highly abstract representation of instances and has different forms in different learning frameworks, such as the mixed Gaussian kernel in Fig. 1(b), the point in Fig. 1(c), and the box mask in Fig. 1(d). The intersection hypothesis assumes that a foreground pixel in the learning objective is shared by multiple instances(Fig. 1(b)), while in Fig. 1(c-e) without the intersection hypothesis, each foreground pixel is only occupied by one instance.
>
> Sorry for any misunderstanding, and we have revised it from *line-72* to *line-73* in the new version.
>
> ---
>
> **Q8:  Visualizing how PML really addresses intersection hypothesis.**
>
> **A8:** The intersection hypothesis is not a problem, but a design principle. The elimination of the intersection hypothesis is implemented in the divide stage, where PML assigns each pixel to only one element in the GT point set. To make this clear, we plot the procedure of PML (the divide stage and conquer stage) in Fig. 6 and explain it in Sec. B of the revised appendix, to demonstrate how the intersection hypothesis is removed in PML.
>
> ---
>
> **Q9: PML on recent SOTA models.**
>
> **A9:**  See cGtL-Q2.
>
> ---
>
> **Q10:  quantitatively showcase how much PML can actually eliminate the intersection hypothesis**
>
> **A10:** The intersection degree can be quantitatively measured by localization performance, as previous studies have shown that training without the intersection hypothesis can improve localization performance. To showcase this quantitatively, we have updated the paper and presented the detailed performance on the NWPU localization benchmark in Sec. J of the Appendix and Tab. 9 (bottom). PML achieves better F1-measure than all previous methods, including those with and without the intersection hypothesis. In different head size levels, PML also achieves the best recalls from A2 to A4 (head area ranges from 1e2 to 1e5) and achieves the second-best recalls at other levels.

---

> > ### Comment · Reviewer_cGtL · 2024-11-25
> >
> > Thanks the authors for the responses. My most concerns have been treated carefully. I maintain my initial rating.
> >
> > Only one remaining comment on the use of PML on SOTA models. In the rebuttal, the authors show that the PML loss outperforms the STEERER model with multi-res density output, which is good. Yet, since PML is considered as a generic loss, it should work compatible with most existing models. What I really want to see is that whether PML can be INCORPORATED into SOTA models to further improve the performance and set a new SOTA.

---

> ### Author Response · Authors · 2024-11-26
> **Response to Reviewer cGtL -- part 3**
>
> Thanks for your comments:
>
> **Q11: incorporate PML into SOTA models to further improve the performance**
>
> **A11:** We tried to incorporate PML into STEERER and conduct experiments on ShTec A/B, the results are like this:
>
> | ShTech A                | MAE | MSE |
> | :-:                 | :-:   | :--:     |
> | STEERER                | 54.5 | 86.9 |
> | STEERER  + PML |  **52.6**| **85.5** |
>
> | ShTech B                | MAE  | MSE  |
> | :-:                 | :--:      | :--:     |
> | STEERER                | 5.8     | 8.5     |
> | STEERER  + PML |  **5.3**    | **8.4**     |
>
> The performance of STEERER with PML is better than the one without PML, but we observed that the training process is unstable and converges slowly: STEERER + PML achieves an MAE smaller than 53 on ShTech A after 1150 training epochs, while the simplified version in our approach achieves similar result in just 300 epochs.
>
> This may be caused by the inherent loss in STEERER, which is specifically designed for pixel-wise L2 loss between prediction and stable training targets (Gaussian kernel with fixed sigma (15 in STEERER)). However, PML is a pixel-to-point loss as shown in (24) and (14). The pixel-wise learning objective presented in (7) derived from PML is not a fixed but a dynamic one changing according to the current prediction. During density map selection in STEERER, the dynamic property makes it difficult to reflect which density map resolution is closest to the ground truth, causing the selective inheritance learning in STEERER to fail. An effective and stable way to incorporate pixel-to-point loss like BL, GL, and our PML into the training framework to select the best density map from various resolutions may be a future work to explore.
>
>
> Besides, our ongoing work is to incooperate PML into the other SOTA, CrowdDiff. However, currently reproducing CrowdDiff is challenging. There is no one has been able to reproduce CrowdDiff's results, as listed in the issues module in [Issues · dylran/crowddiff](https://github.com/dylran/crowddiff/issues).

---

### Official Review · Reviewer_JtJ9 · 2024-11-04

**Soundness:** 3
**Presentation:** 3
**Contribution:** 2
**Rating:** 6
**Confidence:** 4

**Summary:**

The paper introduces Proximal Mapping Loss (PML) to improve crowd counting and localization and challenges the basis of prevailing loss functions used in density regression. PML divides the predicted density map into smaller, non-overlapping point neighbors and is processed independently to construct a target for learning. The paper also links various existing loss functions used in crowd counting to PML, showcasing the generalizability across existing methods.

**Strengths:**

The paper demonstrates limited originality through its approach, which addresses a key limitation, the intersection hypothesis, in existing methods. The method offers a fresh perspective on how crowd density can be modeled. The paper provides good theoretical foundations for the proposed method and empirical results compared to existing methods. The paper is well-structured and progresses coherently to deliver the proposed method.

**Weaknesses:**

While the paper demonstrates that PML outperforms several existing loss functions, it does not explain why PML works better for density-based crowd-counting methods. The method’s reliance on nearest-neighbor assignment for dividing the predicted density map into point-neighbor cases could be problematic, particularly in dense crowd structures. NN algorithms may not capture the correct relationships between nearby points, leading to inaccurate density estimations in such edge cases. The paper does not address how this issue is handled or provide evidence that PML performs well in extreme cases.

**Questions:**

Major Comments

1. There are existing density-based crowd-counting methods that already don’t rely on the intersection hypothesis, such as STEERER and CrowdDiff. STEERER achieves non-overlapping density through scaling, and CrowdDiff uses narrow kernels to prevent overlapping. Both methods use Gaussian kernels and avoid overlapping in density kernels. In that case, why would the non-overlapping nature of PML work better than non-overlapping Gaussian kernels?
2. The paper lacks a comparison against CrowdDiff, the state-of-the-art NWPU crowd-counting benchmark.
3. How does eliminating the intersection hypothesis improve crowd counting and localization performance?
4. Fig. 3 shows the density maps obtained with or without applying PML. If it is the former, how is localization achieved from the density maps? Also, in some cases, more than one density kernel is predicted on a single face. How are these false positives filtered for localization and counting?
5. How does the nearest neighbor approach in the divide and conquer stage impact performance in dense versus sparse crowd scenes?
6. The counting loss was tested using the nearest neighbors in CLTR. How does the PML’s nearest neighbor assignment compare to CLTR?

Minor comments

7. How does PML handle edge cases where crowd members are extremely close together or partially occluded? Especially with partial occlusions, even though the pixel values are not influenced by neighboring objects, the density kernels could have some overlapping as this is a distribution predicted over image coordinates conditioned on pixel values.
8. The detailed results from the NWPU evaluation could help address these edge cases for the crowd-counting community.
9. Missing citation for RAZNet in line 135.

---

> ### Author Response · Authors · 2024-11-22
> **Response to Reviewer JtJ9 -- part 1**
>
> **Q1: Why would the non-overlapping nature of PML work better than non-overlapping Gaussian kernels?**
>
> **A1**: *"Sec 4.1: From Dynamic L2 Loss to Gaussian-Blurred L2 Loss"* theoretically analyzes the differences and connections between PML and the Gaussian-blurred L2 loss.
>
> On the one hand, from *line-245* to *line-250*, a learning objective ${p^*}$ for L2 loss is constructed dynamically according to the current prediction. On the other hand, from *line-257* to *line-266*, we demonstrate how to add conditions to (7) to facilitate the transition from a  **dynamic ${p^\*}$** to a **fixed Gaussian kernel ${p^\*}'$**.
>
> Compared with non-overlapping fixed Gaussian kernels, a dynamic learning objective ${p^*}$ can achieve better performance because it finds a customized and more learnable target for the current model, as explored in numerous studies:
> - [a] introduces *self-correction supervision*, estimating a dynamic learning objective ${p^*}$ based on a Gaussian Mixture Model derived from the prediction ${a}$.
> - [b] proposes ADMG, a *refiner module* that fuses different Gaussian kernels to construct the learning objective ${p^*}$, with the *refiner module* being supervised by the current prediction ${a}$.
> - [c] further lifts the restrictions in [b] and introduces KDMG, which estimates a kernel at each pixel to construct ${p^*}$ according to the ground truth (GT) point map. Similar to [b], the learnable constructor in KDMG is also supervised by the current prediction ${a}$.
> - The loss function in [d] also includes an L2 term (pixel loss), where the learning objective ${p^*}$ is constructed dynamically by aggregating the transport plan between GT and prediction ${a}$, estimated through unbalanced optimal transport.
> - The learning objective of P2PNet, as presented in [e], is also constructed based on GT points and the current estimate ${a}$, which is involved in the formation of the cost matrix (18).
>
> ```
> [a] Adaptive Dilated Network with Self-Correction Supervision for Counting
> [b] Adaptive Density Map Generation for Crowd Counting
> [c] Kernel-based Density Map Generation for Dense Object Counting
> [d] A Generalized Loss Function for Crowd Counting and Localization
> [e] Rethinking Counting and Localization in Crowds: A Purely Point-Based Framework
> ```
>
> These studies have shown that adapting the learning target (e.g., the intermediate representation of the density map) can lead to better counting performance by better fitting the properties of the data. We add related content to line-267 in the revised version.
>
> ---
>
> **Q2:  Dividing the predicted density map into point-neighbor cases could be problematic, particularly in dense crowd structures.**
>
> **A2:** As described from *line-073* to *line-079*, several studies have explored the possibility of dividing the prediction into non-overlapping patches, *i.e.*, LSC-DNN, TopCount, and IIM. Additionally, ikNN and FIDTM also demonstrate that it is reasonable to divide the prediction into irregular patches based on the distance to GT points, of which the description has been added to the introduction (*line-077*) and related works (*line-167*) of the new version.
>
> ---
>
> **Q3:  NN may not capture the correct relationships between nearby points. Provide evidence that PML performs well in extremely nearby points.**
>
> **A3:** In the revised version, we add Sec. C -- "Sparse *vs.* Dense Crowd in PML" and Fig 7 to demonstrate it. According to Fig. 7, PML can predict a count close to 1 in each point-neighbor case even if it is very dense (NN < 4: the distance between the concerned point and its nearest neighbor in GT is smaller than 4).
>
> ---

---

> ### Author Response · Authors · 2024-11-22
> **Response to Reviewer JtJ9 -- part 2**
>
> **Q4:  Comparison against CrowdDiff.**
>
> **A4:** Crowddiff addresses crowd counting through a coarse-to-fine procedure using diffusion models, which requires a denoising process and takes much time to generate the final density map:
>
> |    UCF-QNRF     | MAE  |  MSE  | Time(s) |
> | :-------------: | :--: | :---: | :-----: |
> | CrowdDiff (T=1) | 74.6 | 134.8 | 0.21    |
> | CrowdDiff (T=2) | 71.9 | 130.5 | 0.43    |
> |    PML(ours)    | 73.2 | 127.5 | **0.08**        |
> | CrowdDiff(T=4)  |   **69.0**   |   **125.7**   | 1.36    |
>
> PML can achieve similar performance when the refine time is less than 4. However, ours is significantly faster than CrowdDiff. Although CrowdDiff can achieve better performance than PML by increasing the refine times (T=4 in the paper), PML outperforms it in other metrics:
>
> - **Inference Efficiency:** As shown in the table above, PML is much faster than CrowdDiff.
> - **Flexibility:** PML's novelty lies in the *loss function* and is independent of the network structure. As shown in Table 4, PML can even improve the performance of MCNN, a very old and small model. From MCNN to HRNet, the best performance can be obtained by simply changing the backbone. This flexibility is not possible for a diffusion model like CrowdDiff.
> - **Theoretical Contribution:** We establish the connection between PML and previous loss functions in crowd counting, as shown in Tab.1, which demonstrates that previous loss functions are special cases of PML within the point-neighbor case.
>
> Combining CrowdDiff and PML could be a future work. However, reproducing CrowdDiff is challenging since there are a lot of issues in reproducing CrowdDiff's results, as listed in [Issues · dylran/crowddiff](https://github.com/dylran/crowddiff/issues).
>
> ---
>
> **Q5: How does eliminating the intersection hypothesis improve crowd counting and localization performance?**
>
> **A5:** Eliminating the intersection hypothesis can improve localization performance, which has been demonstrated in many works, *i.e.,* FIDTM, IIM, and P2P. Their results on the NWPU localization benchmark are listed in Tab. 9 (bottom) of the revised version. However, the counting performances of these methods are not as good as those methods that are based on density regression and follow the intersection hypothesis. In our paper, the main contribution to the improvement of counting **and** localization is attributed to PML, which minimizes the difference between prediction and GT via proximal mapping.
>
> ---
>
> **Q6:  How is localization achieved from the density maps in Fig.3?**
>
> **A6:** In *line-463*, we apply **OT-M** to transform the density map into localization information. If the sum of a local region in a density map is greater than 1, two or more pedestrians may be predicted. However, this issue is not related to PML but rather to OT-M. PML aims to ensure that the sum of each point-neighbor case is close to 1, *i.e.,* the term $||P^\top {a} - 1||_1$ in equation (24).
>
> ---
>
> **Q7:  How does the nearest neighbor approach in the divide and conquer stage impact performance in dense versus sparse crowd scenes?**
>
> **A7:** We add a new section (Sec. C) and figure (Fig. 7) in the Appendix to describe the difference. In a sparse crowd, PML can predict a density map similar to a point map. In dense regions, the density map is close to a Gaussian-blurred density map. Note that the nearest neighbor is the KNN with K=1. If K > 1, there will be intersections between adjacent point-neighbor cases, which deviates from the original purpose of eliminating the intersection hypothesis.
>
> ---

---

> ### Author Response · Authors · 2024-11-22
> **Response to Reviewer JtJ9 -- part 3**
>
> **Q8:  How does the PML’s nearest neighbor assignment compare to (3) CLTR?**
>
> **A8:** CLTR proposes the KMO-based matching strategy, in which the KNN distance of each concerned point in its set (predicted point set or GT point set) is computed to compare the context similarity in loss. Since the KMO is also designed for point-based counters, we apply it to P2PNet and the improved P2PNet+  derived from PML in Sec. D of the Appendix. The comparison is listed here:
>
> | Method               | MAE  | MSE  |
> | :------------------: | :--: | :--: |
> | P2PNet               | 52.74 | 85.06 |
> | P2PNet+        | 52.49 | 83.02 |
> | KMO + P2PNet         | 55.75 | 91.70 |
> | KMO + P2PNet+  | 52.47 | **82.92** |
> | PML                  | **52.25** | 83.93 |
>
> The performance of KMO + P2PNet is worse than even P2PNet, but "KMO + P2PNet+" can improve the performance and achieve the lowest MSE. However, PML works the best on MAE, because PML separates the prediction into irregular patches by assigning each pixel to its nearest GT points. This ensures that each region naturally captures the spatial relationship similar to KMO. To better understand the proposed PML, a new figure (Fig. 6)  and section (Sec. B) are added to the revised version to demonstrate the process of the divide and conquer stage in PML.
>
> ---
>
> **Q9:  How does PML handle extremely dense scenes or crowds with partially occluded?**
>
> **A9:** Although NN is applied in the divide stage, the model still follows the scheme of density map regression: the count loss $||M^\top {a} - 1||_1$ forces the model to predict the count in each point-neighbor case close to 1. For extremely dense crowds or crowds full of occlusions, the boundary between pedestrians may not be clear, but the count is forced to be close to 1 in each point-neighbor case.
>
> From *line-750*  to *line-785*, We add a new section (Sec. C in the appendix) and a new figure (Fig.7) to describe how PML performs in different density levels. An overall statistic in Fig. 7(a) demonstrates that the predicted count in each point-neighbor case is close to 1, and an example in Fig. 7(b)-(e) visualizes how the model trained with PML performs in both dense and sparse crowds.
>
> ---
>
> **Q10: Detailed results from the NWPU evaluation.**
>
> **A10:** Thanks for your comments. We have added the detailed comparison of both the NWPU counting and localization benchmarks in Tab. 9 and added the corresponding description in Sec. J. On the counting benchmark, PML achieves similar performance to STEERER, and PML excels in S3, where the crowd size variance is the largest, ranging from 500 to 5000. On the localization benchmark, PML achieves outstanding performance at all levels and is much better than STEERER.
>
> ---
>
> **Q11: Missing citation for RAZNet in line 135.**
>
> **A11:** Thanks for kind reminder, we have revised in the new version, see *line-134*.

---

> > ### Comment · Reviewer_JtJ9 · 2024-11-25
> >
> > The authors have responded to the concerns I raised. I have no more additional questions. I will consider these during my final evaluation.

---

### Author Response · Authors · 2024-11-29
**Looking forward to your further comments!**

Dear reviewers and AC:

Thanks again for all of your constructive suggestions, which have helped us improve the quality and clarity of the paper!
Although we cannot upload a revised PDF, we can post replies on the forum before December 2nd. Please don't hesitate to let us know if we can offer any additional clarifications or address any minor issues, as we would love to convince you of the paper's merits. We appreciate your suggestions. Thanks!

-the authors

---

### Meta-Review · Area_Chair_c1c2 · 2024-12-15

**Metareview:**

This paper proposed a novel density regression loss for crowd counting and localization. The reviewers recognized the method's contributions and comprehensive analysis. Experimental results further validate the effectiveness of the proposed approach. The paper is well-structured. Moreover, it provided good theoretical foundations and achieved SOTA performance. The weakness of this paper is that the proposed loss is incorporated into only a few existing modes.  All reviewers rated the paper as marginally above the acceptance threshold, so I recommended it for acceptance.

**Additional Comments On Reviewer Discussion:**

The concerns raised by four reviewers include reference missing, insufficient ablation studies, lack of comparison with some SOTA methods, efficiency analysis and  robustness analysis to noisy annotations etc. The authors provided comprehensive rebuttals, addressing most concerns of  Reviewer cGtL and all the concerns of the other three reviewers. Finally, one reviewer raised his rating and the other three reviewers maintained their positive ratings.

---

### Decision · Program_Chairs · 2025-01-22

Accept (Poster)